



# Analysis of Sulfate Aerosols over Austria: A Case Study

Camelia Talianu[1, 2] and Petra Seibert[1]

[1]Institute of Meteorology, University of Natural Resources and Life Sciences, Vienna, Austria
[2]National Institute of R&D for Optoelectronics, Magurele, Romania

**Correspondence:** Camelia Talianu (camelia.talianu@boku.ac.at)

**Abstract.** An increase of the sulfate aerosols observed in the period 01 – 06 Apr 2014 over Austria is analyzed using in situ measurements at an Austrian air quality background station, lidar measurements at the closest EARLINET stations around Austria, CAMS near-real-time data and particle dispersion modelling using FLEXPART, a Lagrangian transport model. In-situ measurement of $SO_2$, $PM_{2.5}$, $PM_{10}$ and $O_3$ were performed at the air quality background station Pillersdorf, Austria (EMEP
station AT30, 48°43'N, 15°55'E). A CAMS aerosol mixing ratios analysis for Pillersdorf and the lidar stations Leipzig, Munich, Garmisch, Bucharest indicates the presence of an event of aerosol transport, with sulfate and dust as principal components. For the sulfate layers identified at Pillersdorf from the CAMS analysis, backward and forward trajectory analyses were performed, associating lidar stations to the trajectories. The lidar measurements for the period corresponding to trajectory overpass of associated stations were analyzed, obtaining the aerosol layers, the optical properties and the aerosol types. The potential sources of
transported aerosols were determined for Pillersdorf and the lidar stations using the source-receptor sensitivity computed with FLEXPART, combined with MACCity source inventory. A comparative analysis for Pillersdorf and the trajectory-associated lidar stations showed consistent aerosol layers, optical properties and types, and potential sources. A complex pattern of contributions to sulfate over Austria was found in this paper. For the lower layers (below 2000 m) of sulfate, it was found that the Central Europe was the main source of sulfate. Medium to smaller contributions come from sources in Eastern Europe,
the Northwest Africa and Eastern US. For the middle-altitude layers (between 2000 m and 5000 m), sources from Central Europe (Northern Italy, Serbia, Hungary) contribute with similar emissions. Northwest Africa and Eastern US have also important contributions. For the high-altitude layers (above 5000 m), the main contributions come from Northwest Africa, but sources from Southern and Eastern US contribute also significantly. No contributions from Europe are seen for these layers. The methodology used in this paper can be used as a general tool to correlate measurements at in situ stations and EARLINET
lidar stations around these in situ stations.

# 1   Introduction

Sulfate is one of the major aerosol components for particles with diameter smaller than 2.5 μm ($PM_{2.5}$), and for particles with diameter smaller than 10 μm ($PM_{10}$). Other components of the particulate matter (PM) are: organic carbon (OC), elemental




carbon (EC), nitrate, ammonia, mineral and sea salt. Sulfate normally accounts for about 10% to 30% of PM mass concentration (Stocker et al., 2013). More details about the mass concentration of these aerosol components from various rural and urban sites in Europe are given in the IPCC AR5 report (Stocker et al., 2013). The anthropogenic sulfate is produced mainly by oxidation of sulfur dioxide ($SO_2$), or produced by aqueous phase reactions, where $O_3$ and hydrogen peroxide act as important

5   oxidants (Seinfeld and Pandis, 2006), or by adsorption of $SO_2$ on solid particles and subsequent reaction with adsorbed oxygen; the exact mechanism depends on several atmospheric factors (solar radiation, presence of catalysts, $NO_x$, temperature, relative humidity, etc). The adsorption is an important mechanism of sulfate production in urban atmosphere. Soot (elemental carbon) particles and semiconductor metal oxide particulates from mineral dust (e.g. $Fe_2O_3$, $TiO_2$) are potential surfaces for this process (Dupart et al., 2012). The primary precursor for sulfate in the troposphere is $SO_2$ emitted (Solomon, S., D. Qin,

10   M. Manning, Z. Chen, M. Marquis, 2007) from:

- anthropogenic sources: major contribution from combustion of fossil fuel (about 72%) and small contribution from biomass burning (about 2%),

- natural sources: from dimethyl sulfide (DMS) emissions by marine phytoplankton (about 19%) and from volcano eruptions (about 7%).

Beside chemical processes, $SO_2$ is removed efficiently by dry deposition, while sulfate aerosol is removed from atmosphere by wet deposition (Seinfeld and Pandis, 2006). Tropospheric sulfate, mostly in the accumulation mode, has a lifetime estimated of one week (AeroCom, 2018). The key properties of the sulfate are well defined (Solomon, S., D. Qin, M. Manning, Z. Chen, M. Marquis, 2007). Sulfate particles have a cooling effect by light scattering, they are very hygroscopic and therefore represent active cloud condensation nuclei, and they enhance absorption when deposited as a coating on elemental carbon. As main

component in the aerosols, sulfate can have an important contribution to the aerosol optical depth (AOD).

The purpose of this study is

- to assess the relation between the excess with respect to monthly averaged values observed in the in situ measurements of $SO_2$, $O_3$, $PM_{2.5}$ and $PM_{10}$ at the Austrian air quality background station Pillersdorf, at the beginning of Apr 2014, with tropospheric sulfate aerosols observed in Copernicus Atmosphere Monitoring Service (CAMS) products and aerosols

layers observed in lidar measurements at the closest EARLINET stations around Austria

- to estimate the sulfate aerosols potential sources.

The study was based on the synergy of the remote sensing instruments from European Aerosol Research Lidar Network (EARLINET) (Boesenberg et al., 2003), the ceilometer network of the German Meteorological Service (DWD) and in situ monitors, combined with CAMS products and NATALI aerosol-typing model, and atmospheric transport modeling. The ground-

based remote sensing instruments and the CAMS products (assimilating satellite-based remote sensing data) are used to determine the properties of long-range transported aerosols and their vertical distribution. In-situ measurements of PM and trace gases provide local concentrations at the surface and at specific heights in the troposphere. Details about data collection are given in Sec. 2.1.



The back-trajectories analysis relates the aerosol mass loading changes at a receptor location to spatially-fixed sources, identifying the sources by a source-receptor matrix calculation (Seibert and Frank, 2004). In this paper, the analysis of the trajectories has been performed with FLEXTRA (Stohl et al., 1995) (FLEXTRA, 2018), while the estimation of the potential areas of aerosols' sources has been performed using the Lagrangian transport model FLEXPART (Stohl et al., 2010). A detailed
description of the processing of the collected data and the subsequent analysis is given in Sec. 2.3, while the results and the discussion are presented in Sec. 3.

## 2 Methodology

The optical properties of the aerosol considered in this analysis are: backscatter coefficients, extinction coefficients, volume depolarization ratio, particle depolarization ratio (PDepR), lidar ratio (LR) and Ångström exponent (AE).

In this paper, all times are given as UTC times, in the format HH:mm, H being the hour and m the minutes. The altitudes are given as ground-level altitudes.

Whenever referring to measurements, the geographical name is used as indicator for the station location (e.g. Pillersdorf means Pillersdorf site, Leipzig means Leipzig lidar station).

### 2.1 Data collection

The in situ measurement of $SO_2$, $PM_{2.5}$, $PM_{10}$ and $O_3$ were performed at the air quality background station Pillersdorf, Austria (EMEP station AT30, 48°43'N, 15°55'E) (Umweltbundesamt Austria, 2014) which provides:

- daily mean concentration and the maximum value per day of half-an hour averaged concentrations for $SO_2$

- daily mean concentration for $PM_{2.5}$ and $PM_{10}$

- maximum value per day of hourly averaged concentrations and maximum value per day of 8-hours averaged concentra-
tions for $O_3$

The $SO_2$ measurements are performed with a Thermo Scientific Model 43i $SO_2$ Analyzer, with a detection limit of 0.05 ppb, and a range up to 100 ppm. The $PM_{2.5}$ and $PM_{10}$ measurements are performed with an Optical Particle Counter GRIMM Dust Monitor Model EDM180, with a precision of 0.1 $\mu g\,m^{-3}$. The $O_3$ measurements are performed with a Thermo Environmental Instruments Ozone Analyzer, model TEI 49C, with a detection limit of 0.4 ppb and a range of 0.05 to 200 ppm.
The EARLINET lidar stations (Wandinger et al., 2016) used for this study are Garmisch-Partenkirchen (47.47°N, 11.06°E), Leipzig (51.35°N, 12.43°E) (both stations located in Germany), and Bucharest (44.35°N, 26.03°E, Romania). The two DWD ceilometer stations used are located in Munich (48.20°N, 11.45°E) and Schneefernerhaus (47.42 °N, 10.98°E). The following remote sensing devices are deployed:

- High spectral resolution lidar HSRL (Wandinger et al., 2016), located at Garmisch-Partenkirchen, Germany



- Portable Raman multispectral lidar system Polly[XT] (Engelmann et al., 2016), having 8 channels including one water vapour channel and 2 depolarisation channels, located at Leipzig, Germany

- Raman multispectral lidar system RALI (Belegante et al., 2014), having 7 channels including one water vapour channel and one depolarization channel, located at Bucharest, Romania

— Jenoptik ceilometers CHM15kx (Wiegner and Geiß, 2012) at Munich and Schneefernerhaus, Germany

The measurements were done at the following wavelengths: 355 nm, 532 nm and 1064 nm for the elastic channels, 387 nm and 607 nm for the Raman channels, and 532 nm for the depolarization channel. For HSRL, the 313 nm channel was used. For ceilometers, the 1064 nm channel was used.

The lidar and the ceilometer measurements provide the vertical distributions of aerosols, retrieved from the range corrected

signal (RCS, the preprocessed lidar/ceilometer signal corrected with squared range), and the vertical distributions of aerosol polarization, if the instrument is equiped with a polarization channel.

For the remote sensing sites Leipzig, Munich and Bucharest, the column-integrated AOD measurements for various wavelengths were taken from the AERONET sun/sky photometer measurements, the AERONET instruments being collocated with the lidar stations.

In this paper, products from CAMS, the Copernicus Atmosphere Monitoring Service (CAMS, 2018) of the European Earth Observation programme Copernicus were also used; it provides global reanalysis datasets for the period 2003 – 2012, and global near-real-time (NRT) datasets (Dee et al., 2011) for 2013 to present. These datasets were produced (Benedetti et al., 2009) using 4DVar data assimilation in CY42R1 of ECMWF's Integrated Forecast System (IFS), with 60 hybrid sigma/pressure (model) levels in the vertical, with the top level at 0.1 hPa. Atmospheric data are available on these levels and they are also

interpolated to 25 pressure, 10 potential temperature and 1 potential vorticity level(s). "Surface or single level" data are also available.

For this analysis, the CAMS products for "Model levels" and "Surface level" from NRT "Atmospheric composition" dataset were selected for the times 00:00, 06:00, 12:00 and 18:00 for the analysis data and a step of 3 h for forecast data. The mixing ratios of dust, hydrophilic and hydrophobic black carbon, hydrophilic and hydrophobic organic matter and sulfate were

25 retrieved from the lowest 31 model levels, which covers the tropospheric altitudes; temperature and specific humidity were also retrieved for the same model levels. The logarithm of surface pressure was retrieved from the lowest model level, while the geopotential and the aerosol optical depth (AOD) at 550 nm for total aerosol, black carbon, organic matter, dust and sulfate were retrieved from the surface level.

## 2.2 Aerosol and atmospheric transport modelling

FLEXPART and FLEXTRA models were used in this paper for atmospheric transport modelling.

FLEXPART ("FLEXible PARTicle dispersion model") is a Lagrangian particle dispersion model designed for calculating the long-range and mesoscale transport, diffusion, dry and wet deposition, and radioactive decay of air pollutants from point, line area and volume sources. FLEXPART can be run in forward mode, simulating the transport and dispersion of emissions



from given sources towards receptor points or producing gridded output concentration and deposition, or in backward mode from given receptors to produce source-receptor relationships with respect to a point source or gridded sources.

FLEXTRA is a kinematic trajectory model. It simulates only the transport of air parcels by mean winds, ignoring turbulence and convection, and do not represent concentrations, deposition, etc.

For both models the ECMWF (European Centre for Medium Range Weather Forecasts) Era Interim meteorological fields with a horizontal resolution of 0.5°x 0.5°, the lowest 61 vertical levels (corresponding to pressure levels from surface to 250 hPa) out of the 137 vertical levels, and a temporal resolution of 3 h were used. A sub-domain covering a part of North Hemisphere (175°W – 60°E, 0°N – 90°N), including Europe, a part of the Atlantic Ocean, North America and a part of Africa was extracted as "mother" domain.

For the determination of the aerosol optical properties for sites without lidar measurements, where the aerosol composition is determined from CAMS products, the aerosol model from Ref. (Nicolae et al., 2018) was used, called in the following NATALI aerosol model. Six classes of pure aerosol were considered in this model: continental, continental polluted, dust, marine, smoke, and volcanic. In the model, the optical properties are computed for pure aerosols and for mixtures of two or three pure aerosols at fixed wavelengths 350 nm, 550 nm and 1000 nm with the T-Matrix method using light scattering on

non-spherical particles (Mishchenko et al., 1996) for a log-normal distribution of homogeneous particles. The microphysical parameters (effective radius, standard deviation and complex refractive indices) of the components, needed as input in the model, were taken from the GADS database (Global Aerosol DataSet) (Koepke et al., 1997).

For the comparison with optical properties obtained from lidar measurements, the optical properties computed in the model are re-scaled to the lidar wavelengths (355 nm, 532 nm and 1064 nm) using an AE equal to one, as the values of model and

lidar wavelenghts are very close.

### 2.3 Data processing and analysis

#### 2.3.1 Lidar and ceilometer data processing

The vertical profiles of the backscatter coefficients were determined using the Fernald–Klett method (Fernald, 1984; Klett, 1981) for remote sensing instruments with only elastic channels. For instruments with elastic and Raman channels, the

backscatter and the extinction coefficients were determined using the combined method (Ansmann et al., 1992). The PDepR was computed using the volume depolarization ratio and the backscatter coeficients (Freudenthaler, 2016). The AE is computed from the extinction coefficients for the wavelengths 532 nm and 355 nm.

The LR was computed as the ratio of the extinction coefficient to backscatter coefficient. For ceilometers, lidars with only elastic channels and lidar measurements during the day (when only backscatter coefficients can be retrieved), the value of the

LR was taken from the NATALI aerosol model, which gives an estimate of the LR for 14 aerosol types. The values for 532 nm used in this paper are: 23 ± 10 sr for marine, 40 ± 8 sr for dust, 68 ± 6 sr for continental, 52 ± 2 sr for continental polluted, 53 ± 5 sr for polluted dust, 64 ± 8 sr for smoke and 46 ± 10 sr for mixed dust.





The aerosol layers are identified from the lidar measurements with the methodology described in Ref. (Belegante et al., 2014), applied to the RCS profiles.

The aerosol type is determined from the lidar measurements using the NATALI typing algorithm, described in Ref. (Nicolae et al., 2018).

### 2.3.2 CAMS product processing

The values of the CAMS quantities for a given location were computed by interpolating the gridded CAMS values, using the inverse weighting distance interpolation.

The air density and the altitude specific to the model levels were computed according to CY42R1 from IFS documentation (Benedetti et al., 2009).

### 2.3.3 Data analysis

The concentrations of $SO_2$, $PM_{2.5}$, $PM_{10}$ and $O_3$ measured in situ at the air quality background station Pillersdorf were analyzed for sliding periods of one month, to identify excesses with respect to the measured average values. If a significant excess is identified, the corresponding period is analyzed in detail, using also CAMS products at the in situ station and measurements and CAMS products at the closest lidar stations around the in situ station. For Spring 2014, a period with a significant excess was identified in the time interval 15 Mar – 14 Apr, which is presented in this paper.

The CAMS products are retrieved for the in situ site. The time series of mixing ratios of sulfate, dust, organic matter and total aerosols are then analyzed for the same period as the in situ data. If one of the aerosol components has no significant contribution to the aerosol concentration, this component can be neglected in the subsequent analysis of the aerosol. The time series are also retrieved for the lidar stations around the in situ site.

To assess if the excess is caused by a local event or long- or medium-range transported aerosol is involved, a qualitative analysis of the in situ concentration measurements, the time series of mixing ratios at the in situ station and at the lidar stations around the in situ station is done. If the event is present only at the in situ station, we can assume that it is a local event. If the event is seen at some of the lidar stations around the in situ site, the event has contributions from an aerosol transport event.

The layers for the event at the in situ site are then determined by appplying the gradient method (Belegante et al., 2014) on the altitude profiles of aerosol concentrations. The concentrations are computed by multiplying the mixing ratio and the air density.

A statistical analysis of trajectories is then performed for each layer identified at the in situ site. Three-dimensional kinematic hourly trajectories are computed with the FLEXTRA model, run in backward mode for a transport time of 10 –20 days (typical for long-range transport) and in forward mode for few days for several receptor altitudes between 1500 m and 7000 m. Due to the turbulence in the planetary boundary layer, trajectories below 1500 m are usually not included in the analysis, being mostly local trajectories.



A trajectory is associated with a lidar station if the projection of the trajectory on the Earth surface intersects a $0.5° \times 0.5°$ cell centered on the lidar station location. The altitude of the trajectory and the time the trajectory overpasses the lidar cell are the altitude and time of the FLEXTRA trajectory at the corresponding location.

If a trajectory overpasses a lidar station, the lidar measurements for the overpass time are analyzed. The aerosol layers are identified with the same method (Belegante et al., 2014) as for in situ station, applied to the RCS profiles. The optical properties are computed for each identified aerosol layer, as described in Sec. 2.3. The type of the aerosol is determined from the optical properties using the NATALI typing algorithm. The aerosol concentrations are also computed for each layer, using the method described in Ref. (Mamouri and Ansmann, 2017). For each layer, the sulfate fraction (SF) is computed as the ratio of sulfate concentration to total aerosol concentration.

The layers determined from lidar measurements are then compared with the altitude of the trajectories overpassing the lidar station. If the altitude matches a layer within a reasonable distance, the trajectory is associated with the layer. The matching distance is defined as $2\sigma_{\mathrm{lidar}}$, where $\sigma_{\mathrm{lidar}}$ is the effective spatial resolution of the lidar, tipically of the order of $\sim 60$ m.

The source-receptor sensitivity (SRS) is then computed for each layer identified in the sulfate profile at the in situ station using FLEXPART with sulfate as passive tracer. The release is set to the location of the in situ station, at the altitude determined for that layer and the corresponding event time interval. Sources are considered to be situated between $0 - 100$ m. Wet and dry deposition are taken into account in the computation. Combining the source-receptor sensitivity with emission inventories, the relative distributions of $SO_2$ sources for the corresponding sulfate layer are computed. In this study, the MACCity anthropogenic $SO_2$ emission inventories from the Emissions of atmospheric Compounds & Compilation of Ancillary Data (ECCAD) emission database (Darras et al., 2018) was used.

A cross-check of sulfate concentrations from lidar measurements, CAMS sulfate products and FLEXPART is done for the layers at the lidar stations associated with the layers at the in situ station. One expects the values from the three methods to be in agreement.

The optical properties of the aerosol from each layer at the in situ station are then computed according to Sec. 2.2 and compared with the optical properties of the aerosol from the layers at the lidar stations associated with the layers at the in situ station. The optical properties determined at both sites have to be compatible, up to the changes due to the transport from one site to the other. The compatibility is also cross-checked for the type of aerosols at both stations, where the type is determined using the NATALI aerosol model at the in situ site and the NATALI typing algorithm at the lidar station.

## 3 Results and discussion

### 3.1 Results

The in situ measurements of $SO_2$, $O_3$, $PM_{2.5}$ and $PM_{10}$ concentrations recorded at Pillersdorf for the period 15 Mar – 14 Apr 2014 (Umweltbundesamt Austria, 2014) are shown in Fig. 1, together with the averaged values for this period (dotted line). An excess with respect to the averaged values is observed for all measurements in the period $01 - 06$ Apr: 66% for $SO_2$, 11% for





$O_3$, 90% for $PM_{2.5}$ and $PM_{10}$. If the excess period is excluded from the calculation of the average values, the excess increases to 100% for $SO_2$, 14% for $O_3$, 153% for $PM_{2.5}$ and 143% for $PM_{10}$.

The time series of aerosol mixing ratios from CAMS near-real-time data for Pillersdorf are shown for the same period in Fig. 2 for "total aerosols" (sum of all species defined in CAMS data) (a), for sulfate (b) and for dust (c). One observes a sulfate increase with a peak on 02 Apr, and a second, less pronounced peak on 04 Apr. The aerosol mixture is dominated by dust and sulfate, as can be seen by comparing qualitatively the total, sulfate and dust distributions.

Similar distributions, retrieved from CAMS near-real-time data also, are observed for the lidar stations around Pillersdorf, as shown in Fig. 3 for Munich (a), Leipzig (b) and Bucharest (c). From these distributions, one can infer the presence of an event of sulfate transport over Europe.

The vertical profiles of sulfate, dust and "total aerosol" concentrations are shown in Fig. 4 for Pillersdorf, 02 Apr. The sulfate layers, identified with the gradient method, are shown as grayed area in the same figure.

For 02 Apr, from 0:00 to 12:00, sulfate layers mixed with dust are well defined between 2 km and 3 km, and between 4 km and 6 km. During the day, the layers descend slowly and disperse, such that it mixes with dust and the aerosols from the planetary boundary layer. This can also be seen from the concentration profile of "total aerosol", which also shows a similar structure, indicating a common transport path of sulfate and dust as polluted dust nearby Pillersdorf. The evolution of the sulfate and dust layers during the day is correlated with the increase of $SO_2$ and $PM_{2.5}$ concentrations measured in situ, while the evolution of the dust layers is correlated with the increase of the $PM_{10}$ concentration.

For the layers identified above, the back-trajectories of the aerosols were computed with FLEXTRA, starting from Pillersdorf at the time corresponding to the aerosol profiles for a backward period of 12 days. As mentioned before, trajectories below 1500 m are not computed, due to turbulence in the planetary boundary layer.

For 02 Apr, they are shown in Fig. 5 for 00:00 (a), 06:00 (b), 12:00 (c) and 18:00 (d). From the trajectory analysis, the time and the altitude of the trajectories passing over the lidar stations were determined. The station, the time and the altitude are shown in the lower plots of each sub-figure.

The aerosol layers identified Pillersdorf were transported further. Some of the layers pass over the lidar station from Bucharest. Their trajectories were analyzed running FLEXTRA in forward mode for three days, starting from Pillersorf. Fig. 6 shows the forward-trajectories for 02 Apr, 06:00, which pass over Bucharest lidar station on 03 Apr.

The lidar measurements for the stations overpassed by the trajectories determined from the backward and forward analysis are presented as range corrected signal time series (RCS) in Fig. 7 and Fig. 8 and for the event on 02 Apr in Pillersdorf.

Aerosol layers, their optical properties and the concentration were determined from the lidar measurements following the methodology described in Sec. 2. The layers identified are marked on the corresponding RCS plot.

The association of the layers identified from lidar measurements to the altitude of the backward or forward trajectories over the stations corresponding to the layers identified in Pillersdorf was performed for all eight concentration profiles measured (see Fig. 4 for 02 Apr and Fig. 13 for 04 Apr). The association for trajectories from Apr 02, 06:00 is presented in Table 1. The trajectory altitude (Traj. alt.) in the table represents the altitude of the trajectory when overpassing the lidar station. The corresponding layers are also marked in the RCS plots (red line box).





The source-receptor sensitivity was computed for each layer identified in the sulfate profiles at Pillersdorf; the column-integrated source-receptor sensitivity was also computed. Fig. 9 shows the corresponding distributions for the layers L1 (a), L2 (b), L3 (c) and total column (d) from 02 Apr, 06:00.

For each layer, the relative distribution of the $SO_2$ sources was computed from the source-receptor sensitivity and the source inventory MACCity. Fig. 10 (a) shows the distribution for layer L1 at Pillersdorf, 02 Apr, 06:00, while Fig. 10 (b) shows the distribution for the corresponding layer at Leipzig, 31 Mar, 18:00. To evaluate the local distribution of sources near Pillersdorf, a zoomed view of the $SO_2$ relative distribution in shown in Fig. 10 (c) for the sub-domain covering a part of the Europe, centred in Austria (10°W – 40°E, 35°N – 60°N). Similar distributions are shown for layer L2 in Pillersdorf in Fig. 11 (a), with a corresponding layer in Leipzig (31 Mar, 23:00), shown in Fig. 11 (b), and the zoomed view for Pillersdorf in Fig. 11 (c). For layer L3 at Pillersdorf, the distribution is shown in Fig. 12 (a), with associated layers in Munich (Apr 01, 05:00), Garmisch (Apr 01, 14:00 – not shown as very close to Munich) and Bucharest (Apr 03, 13:00) shown in Fig. 12 (b) and Fig. 12 (c), respectively, and a zoomed view for Pillersdorf in Fig. 12 (d).

For the lidar stations, a comparison of concentrations computed from the lidar measurements with the sulfate concentrations computed from CAMS values for the lidar station location and the concentrations computed from the modelled SRS are given in Table 2.

The optical properties, the sulfate fraction and the aerosol types for the aerosol layers identified for Pillersdorf, 02 Apr, 06:00 and the associated layers at the lidar stations are given in Table 3. For Leipzig and Bucharest, the optical properties are computed from the lidar measurements; for Pillersdorf, Garmisch and Munchen they are computed using the NATALI model.

The peak on 04 Apr was also analyzed similarly to the peak on 02 Apr. The corresponding vertical profiles of sulfate, dust and "total aerosol" concentrations are shown in Fig. 13. From the backward and forward trajectory analysis, only one lidar station could be associated with a trajectory, for layer L2 at Pillersdorf, 12:00. The corresponding RCS at the lidar station is shown in Fig. 14. The SRS for the identified layers at Pillersdorf, 12:00, are presented in Fig. 15. Layers at Pillersdorf were associated to layers at the lidar stations; they are given in Table 4. The comparison of the aerosol concentrations at the lidar station over-passed is given in Table 5, and the optical properties are given in Table 6.

## 3.2 Discussion of the results

The daily variations of the in situ measurements of $SO_2$, $O_3$, $PM_{2.5}$ and $PM_{10}$ concentrations depend on more factors, such as variations in source emissions, photochemical reactions, meteorological conditions, PBL heights and short-, medium- and long-range transport of aerosols.

Fig. 1 indicates a period between 27 Mar and 6 Apr in which in situ measurements of $SO_2$, $O_3$, $PM_{2.5}$, $PM_{10}$ concentrations recorded at Pillersdorf exceed the averaged values for the period 15 Mar – 14 Apr 2014. A significant load of aerosols in the atmosphere in this period is also confirmed by the AOD values between 0.07 and 0.73 for Pillersdorf, retrieved from CAMS products, which are above the AOD threshold of 0.06 for clear atmosphere (Kaskaoutis et al., 2012). For the period 27 Mar to 31 Mar, no significant load of aerosols is observed at the lidar stations around Pillersdorf, therefore no medium- or long range transport of aerosols is involved. The source-receptor sensitivity computed for 31 Mar (not shown in this paper) points to a





short-range transported event, of small duration and at low altitude, with sources in the South Eastern of Austria. This event is
not described in this paper.

From a qualitative analysis of in situ concentrations for $PM_{10}$, $PM_{2.5}$ and $SO_2$ (Fig. 1) and the CAMS time series of mixing
ratios for dust, sulfate and total aerosol at Pillersdorf (Fig. 2) and the lidar stations around Pillersdorf (Fig. 3), the presence of
an event of sulfate transport over Europe can be inferred, with two peaks, on 02 Apr and 04 Apr, respectively.

On 02 Apr, one observes from the concentration profiles (Fig. 4) that in the morning the dust was dominant in the layer
between 0.55 km and 1.50 km and in the layer between 1.98 km and 3.11 km, while sulfate was dominant in the higher
altitude layer, between 4.20 km and 6.15 km. In the afternoon, the sulfate concentration increases gradually in the lower
layers, mixing with the dust, while the upper layer become thinner (layer range from 4.0 km to 5.0 km).

The back-trajectories for 02 Apr (Fig. 5) show a consistent pattern. In the morning (00:00 and 06:00), the lower trajectories
(below 2000 m) originate from Eastern and Southern United States (US), transverse the North Atlantic Ocean and pass over
Central Europe, spending ∼ 6 days in this region, arriving at Pillersdorf from the northwest direction. The middle-altitude
trajectories (2000 m – 5000 m) originate from Southern US, transverse the ocean and pass over Northwest Africa (spending ∼ 3
days), arriving in the Central Europe from southwest, then arriving along the Alps at Pillersdorf. The high-altitude trajectories
(above 5000 m) transverse the ocean, arriving at Pillersdorf from the west direction. In the afternoon (12:00 and 18:00), the
lower trajectories originate from Eastern Europe, while the middle-altitude and high-altitude trajectories originate from Eastern
US, transverse the ocean and the Northwest Africa, arriving at Pillersdorf from the west direction.

The SRS patterns, shown in Fig. 9, and the relative distributions of $SO_2$, shown in Fig. 10, Fig. 11 and Fig. 12, indicate
the influence of five source regions for the transport of the sulfate event recorded on day 02 Apr at Pillersdorf: Southern and
Eastern US, Northwest Africa, Central Europe and Eastern Europe.

For the lower layers, the Central Europe, including industrial centres from the "Black triangle" (Eastern Germany, Southwest
Poland and Czech Republic) was the main source contributing to sulfate transported over Northern Austria, where Pillersdorf
station is situated. Medium to smaller contributions come from sources in Eastern Europe, the Northwest Africa and Eastern
US.

For the middle-altitude layers, sources from Central Europe (Northern Italy, Serbia, Hungary) contribute with similar emis-
sions. Northwest Africa and Eastern US have also important contributions.

For the high-altitude layers, the main contributions come from Northwest Africa, but sources from Southern and Eastern US
contribute also significantly. No contributions from Europe are seen for these layers.

For the peak on 04 Apr, having only one lidar station associated to aerosol trajectories, the analysis is more difficult. From
the existing information, we can conclude that the pattern is similar with layer L2 and L3 from 02 Apr, with contributions from
Northen Italy, Northwest Africa and Southern US.

The AEs for the event have values between 0.67 and 0.79, which correspond to a mixture of fine and coarse particles, with
size distribution centered on 0.75 μm. For this size distribution, the sulfate (Ding et al., 2017) and the dust (accumulation mode)
are the dominant aerosols. The LR is comparable for all sites, having values between 45 and 55 sr, while the linear PDepR has





values between 0.07 to 0.22. These values correspond to low to medium absorbing aerosol with non-spherical shape (Nicolae et al., 2018).

The aerosol type is determined from the optical properties for the layers identified in this event, at the in situ station and the lidar stations. Consistent aerosol type was found between the in situ station and the lidar stations along the trajectories. The

5 changes in the values of the aerosol LR, AE and linear PDepR along the trajectories can be explained by:

- the mixing of dust with secondary sulfate from anthropogenic sources during the transport paths to Leipzig, Munich, Garmisch-Partenkirchen, Pillersdorf and Bucharest

- the adsorption of the $SO_2$ on the dust mineral oxides compounds. The sulfate particles are expected to be formed by $SO_2$ oxidising on dust surface due to mineral oxides compounds from dust (e.g. hematite).

Monthly-averaged maps of column mass density for sulfate are available from EarthData NASA GIOVANNI (NASA, 2018) online data system. For short-time events, they can be used only for a qualitative interpretation, being monthly averaged. The map of sulfate column mass density for Mar 2014, Fig. 16 (a), shows an increased density over South–Eastern and Eastern US and a reduced density over Central Europe. For Apr 2014, shown in Fig. 16 (b), the density increases over Central Europe.

## 4 Conclusions

The excess of $SO_2$, $PM_{2.5}$, $PM_{10}$ and $O_3$ observed in the period 01 – 06 Apr 2014 at the Austrian air quality background station Pillersdorf was analyzed using in situ data, lidar measurements at the closest EARLINET stations around the in situ site, CAMS near-real-time data, and aerosol and atmospheric transport modelling. This excess was associated with the transport of sulfate aerosols, mixed during the transport with dust. By correlating the local information with a trajectory analysis and an analysis of aerosol potential sources, a complex pattern of contributions to sulfate at the in situ station was found. The lower

layers (below 2000 m) originated mainly from the Central Europe. Medium to smaller contributions came from sources in Eastern Europe, the Northwest Africa and Eastern US. For the middle-altitude layers (between 2000 m and 5000 m), sources from Central Europe (Northern Italy, Serbia, Hungary) contributed with similar emissions. Northwest Africa and Eastern US have also important contributions. The high-altitude layers (above 5000 m) originated from sources from Northwest Africa and from Southern and Eastern US, as transported secondary sulfate mixed with dust. The effect of medium- and long-range

transport of aerosol is significant, and can not be neglected when analyzing the air quality at an in situ station. For a quantitative analysis and modelling of aerosol deposition, more measurements are needed, including precise vertical aerosol profiles at the in situ station.

The methodology developed in this paper allows to obtain a better understanding of the effects of aerosol transport on the in situ measurements. It can be used as a general tool to correlate measurements at in situ stations with ground-based remote

sensing stations located around these in situ stations. A dedicated paper for the methodology, extended to trace gases and other aerosols, with analysis of more case studies is under preparation.



*Author contributions.* CT collected and processed all data, developed the methodology and performed the data analysis. Both authors contributed to the optimization of the analysis and the interpretation of the results. PS provided the pre-release of FLEXPART version 10, with a better wet deposition and other improvements. The manuscript was prepared by CT with contributions from PS.

*Competing interests.* The authors declare that they have no conflict of interest.

5  *Acknowledgements.* This study was supported by the Austrian Science Fund FWF, Project M 2031, Meitner-Programm. We thank the Principal Investigators and their staff for establishing and maintaining the EARLINET lidar sites, the DWD ceilometers and the AERONET stations. We thank the staff from the Environment Agency Austria who provided the in situ data. We acknowledge ECCAD and CAMS for making data accessible and providing tools for data analysis. The sulfate column mass density maps were produced with the Giovanni online data system, developed and maintained by the NASA GES DISC. We also acknowledge the mission scientists and Principal Investigators
10  who provided the data used in this research effort.



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





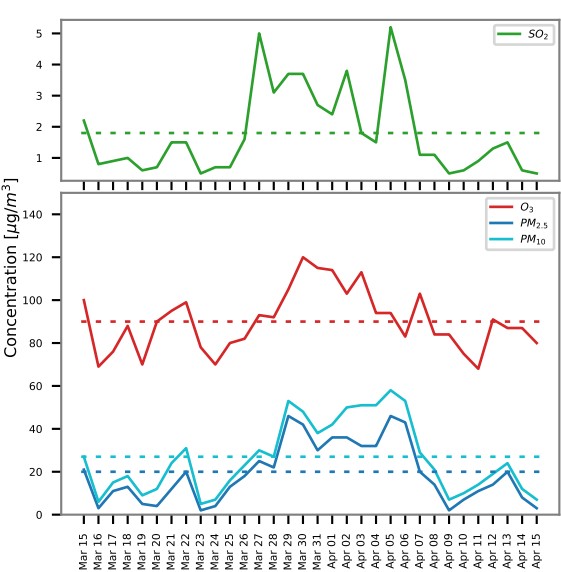

**Figure 1.** In-situ $SO_2$, $O_3$, $PM_{2.5}$ and $PM_{10}$ concentrations measured at Pillersdorf, Austria (EMEP station AT30, 48°43'N, 15°55'E). The dotted lines represent the averaged values for the plotted period.





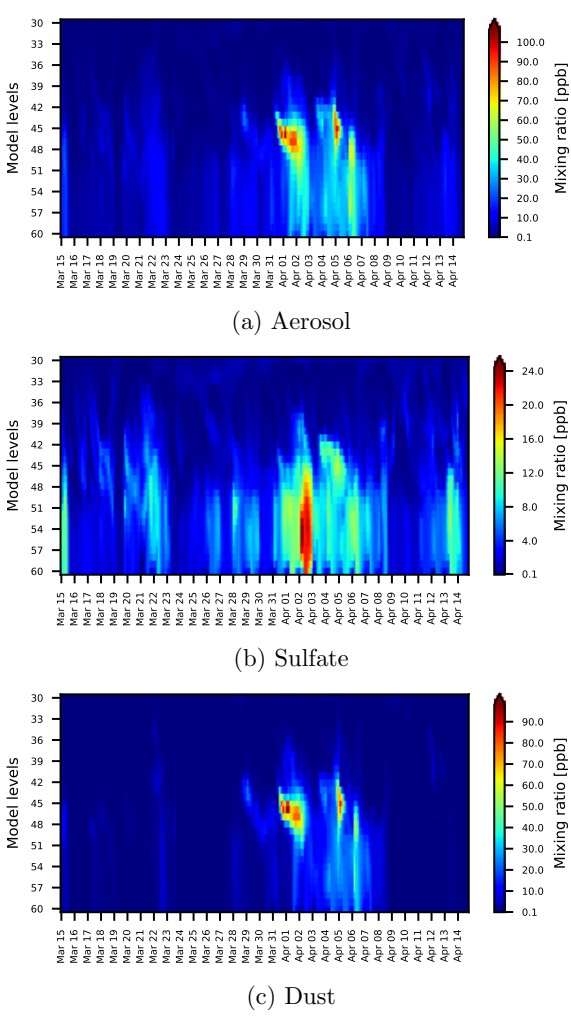

(a) Aerosol

(b) Sulfate

(c) Dust

**Figure 2.** Time series of CAMS mixing ratios for total aerosol (a), sulfate (b) and dust (c), Pillersdorf, 15 Mar – 14 Apr 2014.





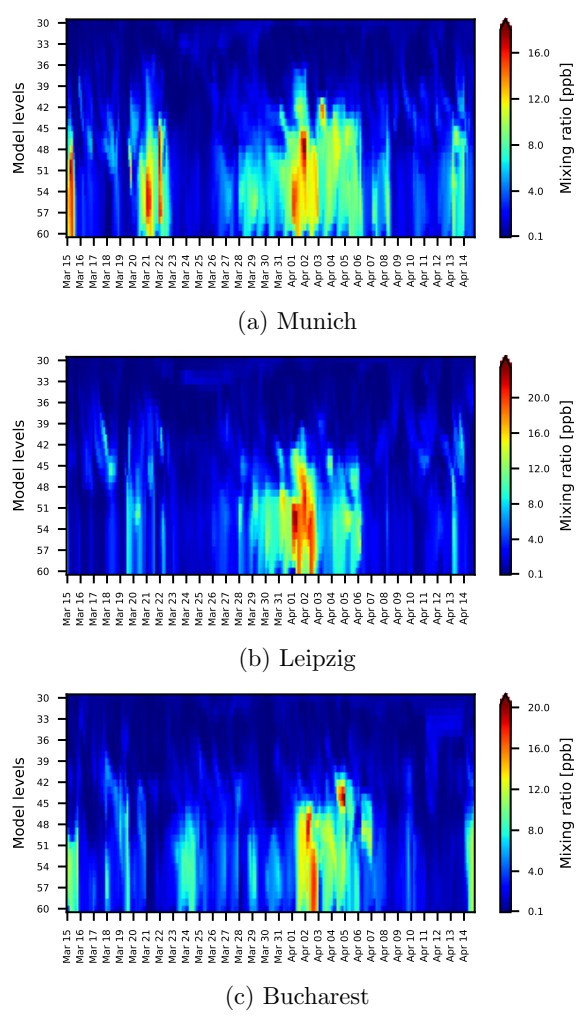

**Figure 3.** Time series of CAMS mixing ratios for sulfate for Munich (a), Leipzig (b) and Bucharest (c), 15 Mar – 14 Apr 2014.





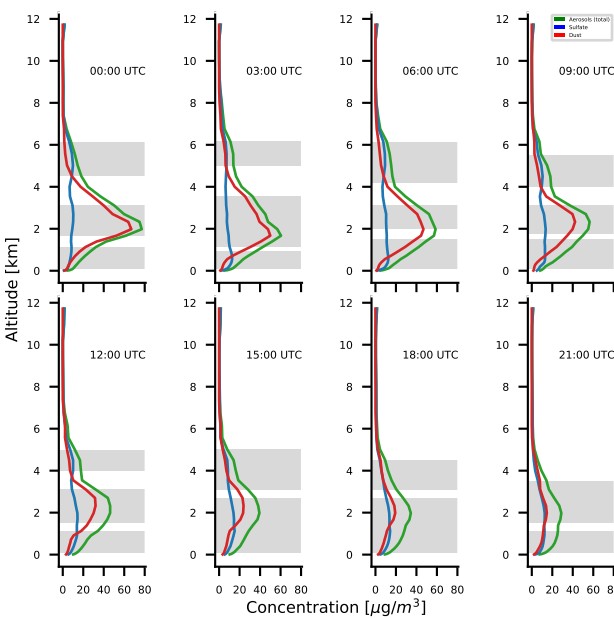

**Figure 4.** CAMS aerosol, sulfate and dust profiles for 02 Apr 2014, Pillersdorf. Grayed area represents the identified sulfate layers.



(a) 00:00

(b) 06:00

(c) 12:00

(d) 18:00

**Figure 5.** Pattern of back-trajectories (upper plot of sub-figure) and their altitude profile, including overpassed lidar stations (lower plot of sub-figure) for Pillersdorf, 02 Apr 2014 at 00:00 (a), 06:00 (b), 12:00 (c), 18:00 (d).





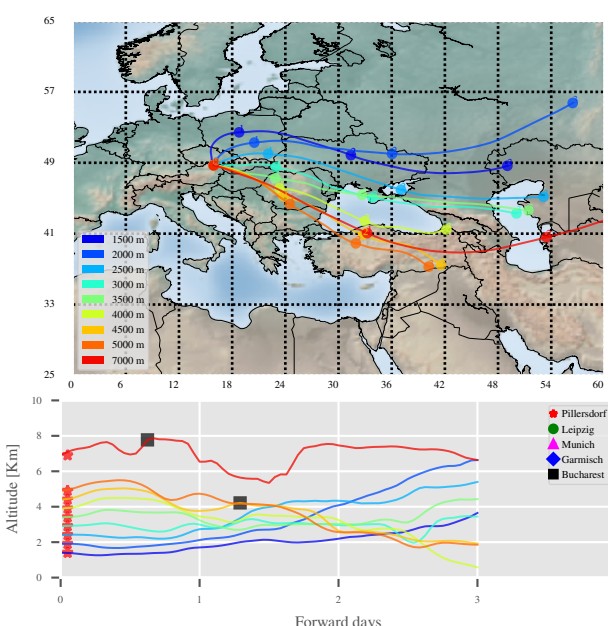

**Figure 6.** Pattern of forward-trajectories (upper plot) and their altitude profile, including overpassed lidar stations (lower plot) for Pillersdorf, 02 Apr 2014, 06:00.





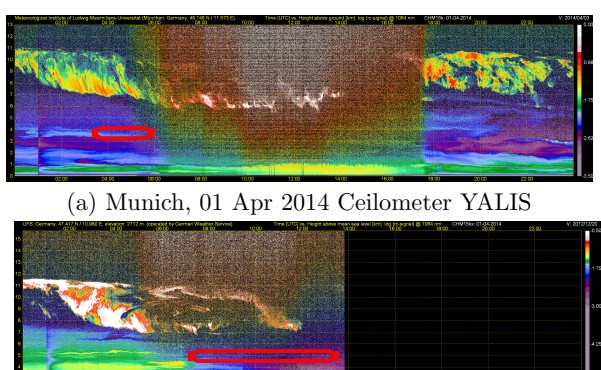

(a) Munich, 01 Apr 2014 Ceilometer YALIS

(b) Garmisch, 01 Apr 2014 Ceilometer

**Figure 7.** log(range corrected signal) at 1064 nm, 24 h, for Munich (a) and Garmisch (b) stations. The red line boxes represent the identified layers.





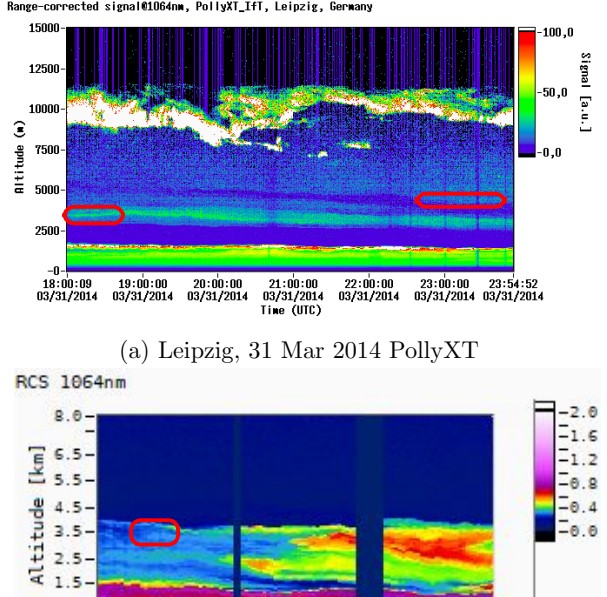

(a) Leipzig, 31 Mar 2014 PollyXT

(b) Bucharest, 03 Apr 2014 RALI

**Figure 8.** Range corrected signal at 1064 nm for Leipzig (a) and Bucharest (b) stations. The red line boxes represent the identified layers.





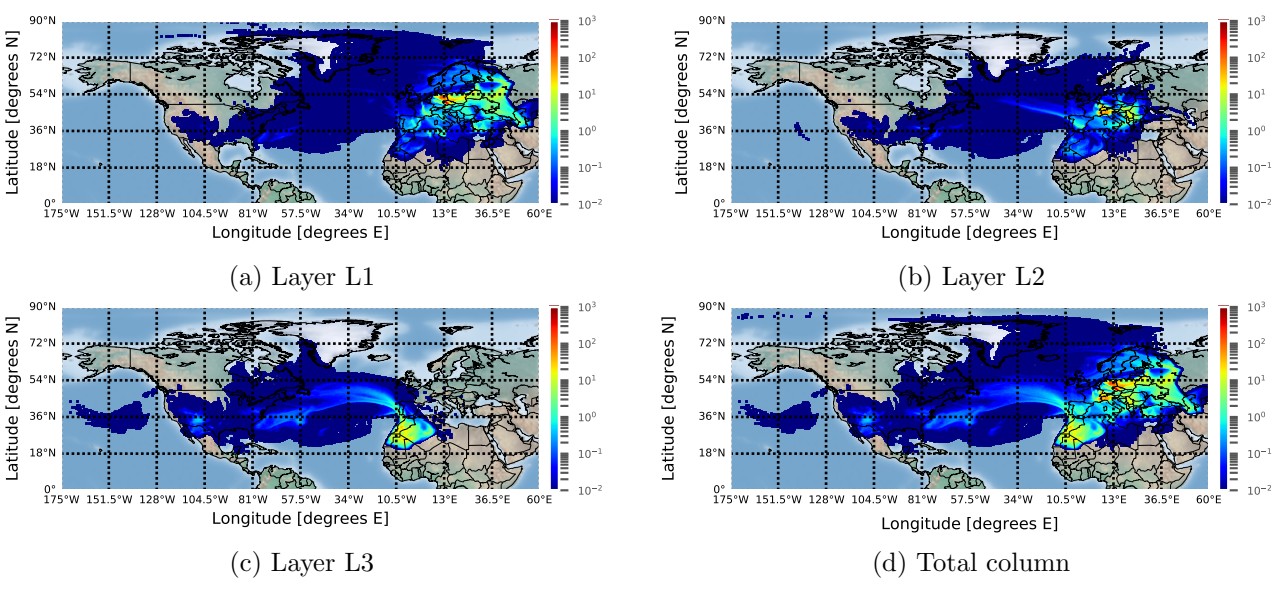

(a) Layer L1        (b) Layer L2

(c) Layer L3        (d) Total column

**Figure 9.** Source-receptor sensitivity for layer L1 (a), L2 (b) and L3 (c) and total column (d), Pillersdorf, 02 Apr, 6:00



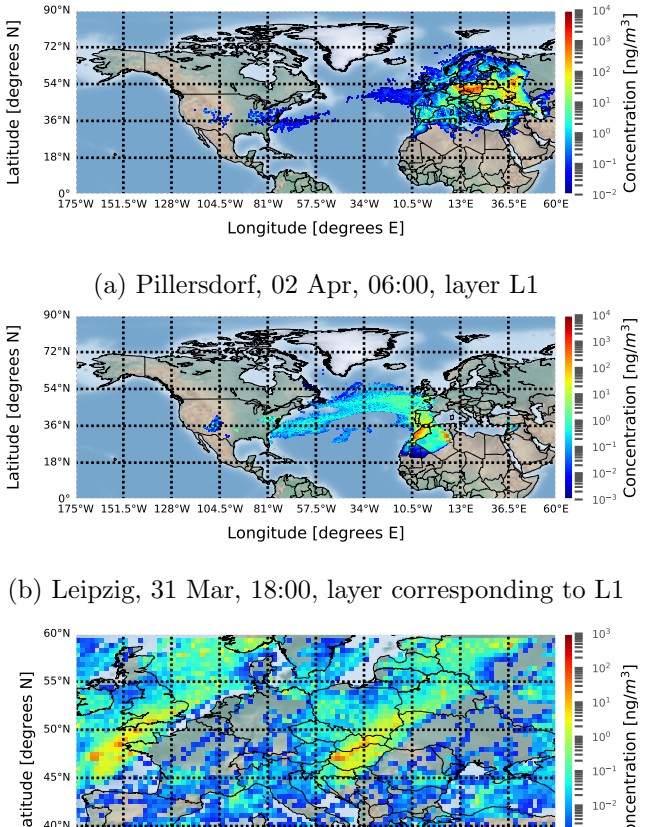

(a) Pillersdorf, 02 Apr, 06:00, layer L1

(b) Leipzig, 31 Mar, 18:00, layer corresponding to L1

(c) Pillersdorf, 02 Apr, 06:00, layer L1, zoomed

**Figure 10.** Relative distributions of SO$_2$ sources for Pillersdorf layer L1 (a), Leipzig (b); zoomed distribution for Pillersdorf layer L1 (c).





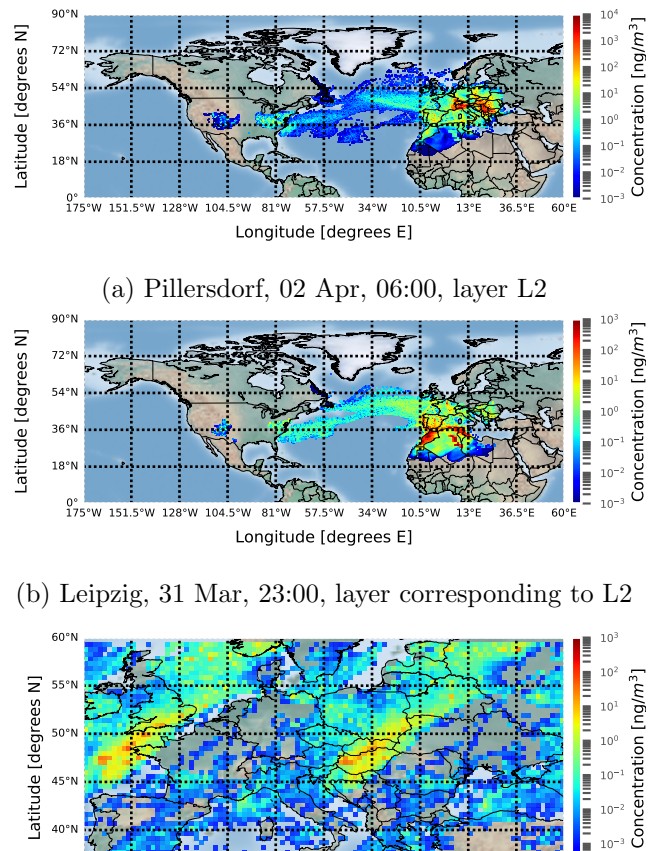

(a) Pillersdorf, 02 Apr, 06:00, layer L2

(b) Leipzig, 31 Mar, 23:00, layer corresponding to L2

(c) Pillersdorf, 02 Apr, 06:00, layer L2, zoomed

**Figure 11.** Relative distributions of $SO_2$ sources for Pillersdorf layer L2 (a), Leipzig (b); zoomed distribution for Pillersdorf layer L2 (c).





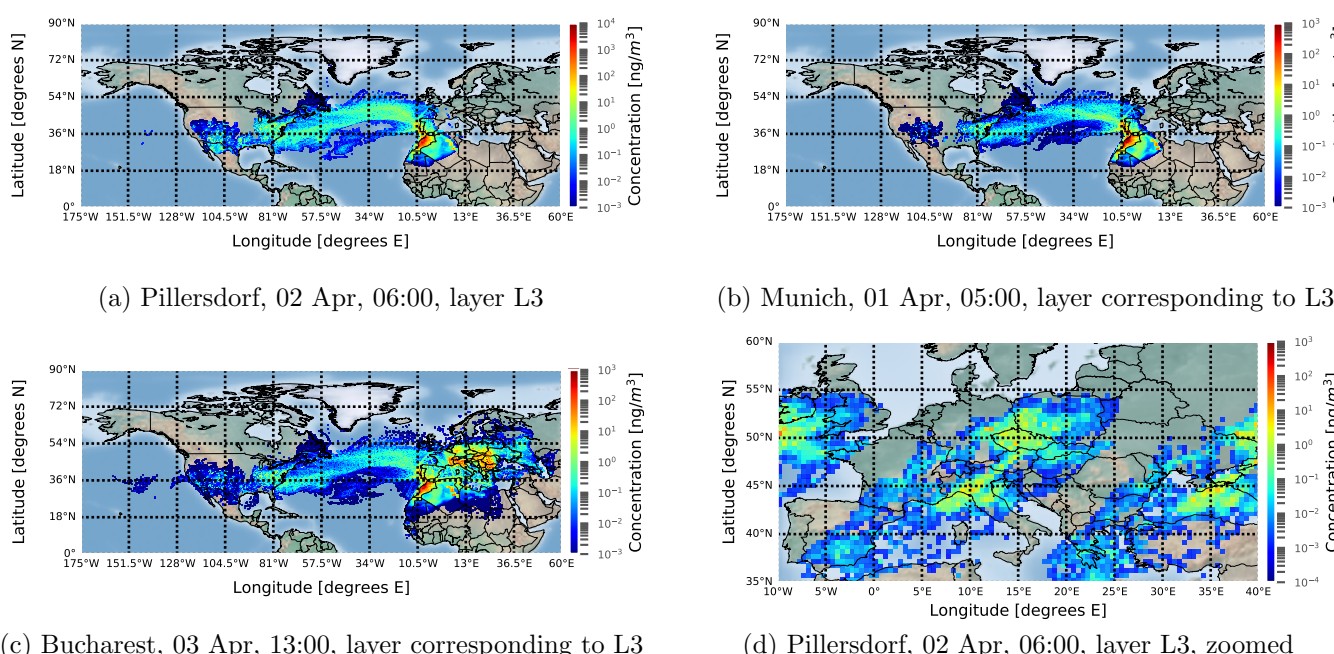

(a) Pillersdorf, 02 Apr, 06:00, layer L3

(b) Munich, 01 Apr, 05:00, layer corresponding to L3

(c) Bucharest, 03 Apr, 13:00, layer corresponding to L3

(d) Pillersdorf, 02 Apr, 06:00, layer L3, zoomed

**Figure 12.** Relative distributions of SO$_2$ sources for Pillersdorf layer L3 (a), Munich (b), Bucharest (c); zoomed distribution for Pillersdorf layer L3 (d).





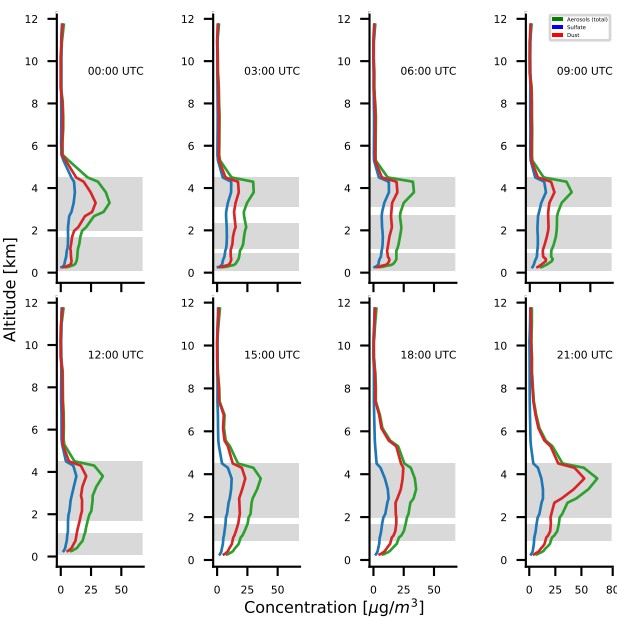

**Figure 13.** CAMS aerosol, sulfate and dust profiles for 04 Apr 2014, Pillersdorf. Grayed area represents the identified sulfate layers.




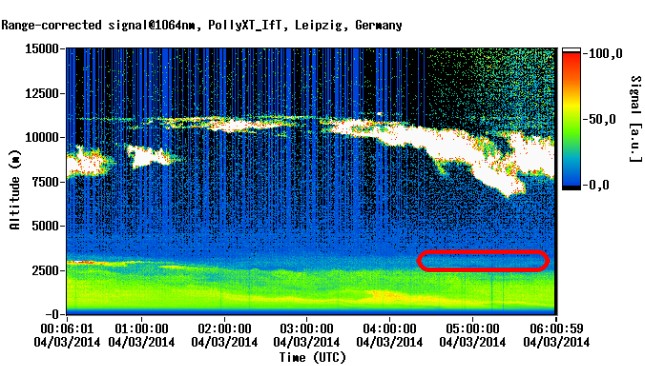

**Figure 14.** Range corrected signal at 1064 nm for Leipzig station, 03 Apr 2014. The red line box represents the identified layer.



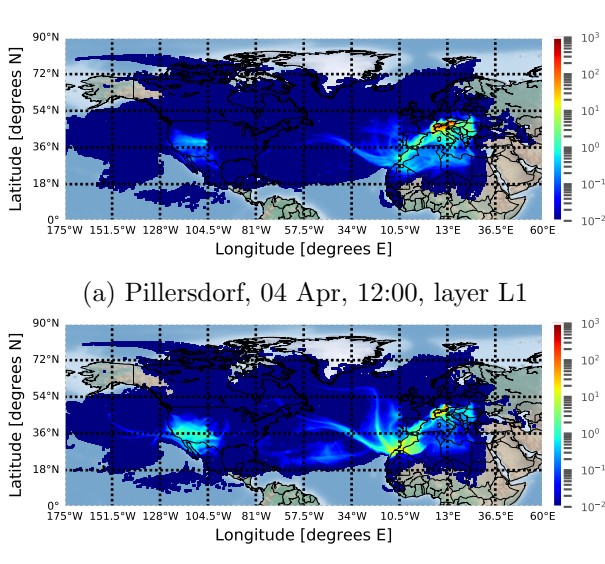

(a) Pillersdorf, 04 Apr, 12:00, layer L1

(b) Pillersdorf, 04 Apr, 12:00, layer L2

**Figure 15.** Source-receptor sensitivity for layer L1 (a) and L2 (b), Pillersdorf, 04 Apr, 12:00



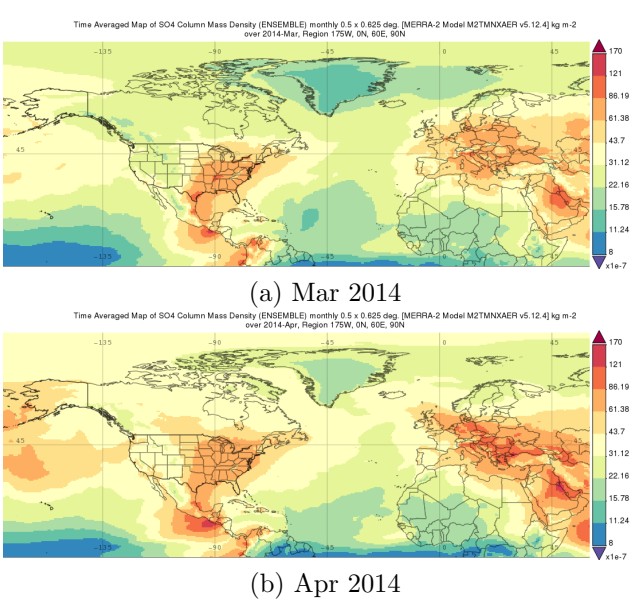

(a) Mar 2014

(b) Apr 2014

**Figure 16.** GIOVANNI time averaged map of sulphate column mass density for Mar 2014 (a) and Apr 2014 (b).





**Table 1.** Association of layers from lidar measurements with layers and trajectories computed for Pillersdorf, 02 Apr 2014, 06:00.

| Pillersdorf | Lidar station, time | |
| --- | --- | --- |
| | Traj. alt. | Lidar layer |
| L1: 0.55 – 1.50 km | Leipzig, Mar 31, 18:00 | |
| | 2.66 km | 2.70 – 3.75 km |
| L2: 1.98 – 3.11 km | Leipzig, Mar 31, 23:00 | |
| | 3.75 km | 3.85 – 4.20 km |
| L3: 4.20 – 6.15 km | Munich, Apr 01, 05:00 | |
| | 4.20 km | 3.54 – 4.43 km |
| | Garmisch, Apr 01, 14:00 | |
| | 4.84 km | 4.91 – 5.81 km |
| | Bucharest, Apr 03, 13:00 | |
| | 3.90 km | 2.70 – 4.05 km |





**Table 2.** Comparison of sulfate concentration computed from lidar measurements, CAMS products and FLEXPART for layers at lidar stations associated with layers from Pillersdorf, 02 Apr 2014, 06:00.

| Layer | $C_{lidar}$ $[\mu g\,m^{-3}]$ | $C_{cams}$ $[\mu g\,m^{-3}]$ | $C_{flexpart}$ $[\mu g\,m^{-3}]$ |
|---|---|---|---|
| Leipzig, Mar 31, 18:00 2.70 – 3.75 km | 14.61 | 12.52 | 12.94 |
| Leipzig, Mar 31, 23:00 3.85 – 4.20 km | 15.96 | 13.48 | 13.42 |
| Bucharest, Apr 03, 13:00 2.70 – 4.05 km | 15.24 | 11.95 | 13.26 |
| Munich, Apr 01, 05:00 3.54 – 4.43 km | 20.14 | 19.58 | 18.98 |
| Garmisch, Apr 01, 14:00 4.91 – 5.81 km | 17.93 | 16.76 | 15.39 |





**Table 3.** Optical properties, sulfate fraction and aerosol types for aerosol layers corresponding to Pillersdorf, 02 Apr 2014, 06:00.

| Layer | LR [sr] | PDepR | AE | SF | Type |
|---|---|---|---|---|---|
| Pillersdorf Apr 02, 06:00 0.55 – 1.50 km | 51 | 0.22 | 0.67 | 0.49 | Polluted dust |
| Pillersdorf Apr 02, 06:00 1.98 – 3.11 km | 55 | 0.10 | 0.76 | 0.33 | Mixed dust |
| Pillersdorf Apr 02, 06:00 4.20 – 6.15 km | 54 | 0.07 | 0.74 | 0.62 | Mixed dust |
| Leipzig Mar 31, 18:00 2.70 – 3.75 km | 55 | 0.20 | 0.79 | 0.25 | Polluted dust |
| Leipzig Mar 31, 23:00 3.85 – 4.20 km | 54 | 0.17 | 0.79 | 0.44 | Mixed dust |
| Bucharest Apr 03, 13:00 2.70 – 4.05 km | 54 | 0.14 | 0.71 | 0.55 | Mixed dust |
| Munich Apr 01, 05:00 3.54 – 4.43 km | 47 | 0.18 | 0.75 | 0.40 | Mixed dust |
| Garmisch Apr 01, 14:00 4.91 – 5.81 km | 45 | 0.16 | 0.71 | 0.41 | Mixed dust |





**Table 4.** Association of layers from lidar measurements with layers and trajectories computed for Pillersdorf, 04 Apr 2014, 12:00.

| Pillersdorf | Lidar station, time | |
|---|---|---|
| | Traj. alt. | Lidar layer |
| L1: 1.98 – 4.50 km | Leipzig, Apr 03, 05:00 | |
| | 2.96 km | 2.70 – 3.45 km |



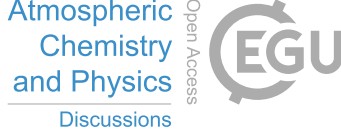

**Table 5.** Comparison of sulfate concentration computed from lidar measurements, CAMS data and FLEXPART for layers at lidar stations associated with layers from Pillersdorf, 04 Apr 2014, 12:00.

| Layer | $C_{lidar}$ [$\mu g\,m^{-3}$] | $C_{cams}$ [$\mu g\,m^{-3}$] | $C_{flexpart}$ [$\mu g\,m^{-3}$] |
|---|---|---|---|
| Leipzig, Apr 03, 12:00 2.70 – 3.45 km | 8.38 | 6.75 | 7.99 |

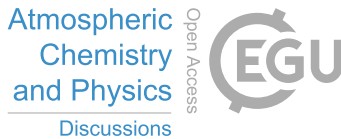
Atmospheric Chemistry and Physics Discussions — Open Access — EGU

**Table 6.** Optical properties, sulfate fraction and aerosol types for aerosol layers corresponding to Pillersdorf, 04 Apr 2014, 12:00.

| Layer | LR [sr] | PDepR | AE | SF | Type |
|---|---|---|---|---|---|
| Pillersdorf Apr 04, 12:00 0.55 – 1.50 km | 54 | 0.07 | 0.75 | 0.25 | Mixed dust |
| Pillersdorf Apr 04, 12:00 1.98 – 4.50 km | 54 | 0.07 | 0.74 | 0.33 | Mixed dust |
| Leipzig Apr 03, 05:00 2.70 – 3.45 km | 55 | 0.11 | 0.76 | 0.74 | Mixed dust |