# Peer review of "Analysis of Sulfate Aerosols over Austria: A Case Study"

_Atmospheric Chemistry and Physics, 2018_

## Referee Comment (RC1) · Anonymous Referee #1 · 14 Jan 2019

General Comments: (overall quality)

The paper is interesting, as it integrates various types of measurements across Europe to demonstrate the impact of possible emission sources on different aerosol concentrations at various ground-level and remote sensing (especially Lidar) monitoring points. Although the 6-day observation period is relatively short, the deliberations certainly demonstrate the potential of FLEXTRA and FLEXPART for short- and possibly also long-range source apportionment studies.

Specific Comments:

Section 2 (Methodology) is rather lengthy and very detailed. Some parts could perhaps be moved into an Appendix Section.

[Figure]

The authors should avoid to mention Trade Names or direct references to companies and commercial instruments, unless absolutely necessary for the understanding of the methods deployed.

Since the trajectories in Fig.5 indicate sources from almost all over Europe (understandable, especially in the lower & mid levels), but also very distant sources (mostly in elevated layers), the authors should show the relevant meteorological maps for the study period (850, 700, 500, 300 & 200 or 250 hPa circulation) to provide physical evidence for "conflicting" circulations in some of the layers, and especially for the "outlying regions". Of course, FLEXTRA ingests the upper air data from ECMWF, but a cross-verification with "real meteorological data" will make the cases more convincing.

P10 / L27 & 28 - "No contributions from Europe are seen for these layers." This may be true for the period in April, as there may not have been any deep convection. However, it would be interesting to also study a summer period with strong convective activity over Central Europe (obviously, in a separate paper !). I am still a bit skeptical about the long-range transport of pollutants - there would be a significant dilution factor . . .! Unless there are major sources emitting ? An indication of such sources would make your findings more convincing.

Technical corrections:

Figs. 4 & 13 - Key for variables needs to be enlarged P8 / L20 - Height of layers "amsl" or "AGL" (also in tables)

---

## Referee Comment (RC2) · Anonymous Referee #2 · 16 Jan 2019

General comments: the manuscript investigates potential non-local sources of sufate aerosols influencing a background air quality monitoring station in Austria, on a case study basis. It integrates different aerosol data platforms and transport models: in situ measurements, Lidar measurements, assimilated satellite-based remote sensing data (CAMS products), and particle dispersion modelling (FLEXPART). This makes the manuscript interesting, although the integration of different data platforms is not new. The results are somehow qualitative, and the manuscript would gain significance if the discussion could be improved to provide some quantitative results. For example, estimating percent contribution of sulfate source regions. In addition, aerosol aging and dilution along the transport could be explored. Uncertainties and limitations of the method could also be better discussed. In my opinion, the discussion should be
improved, according to the specific comments below.

Specific comments:

1) Page 2, Lines 1-2 (Introduction): worldwide in situ observations of refractory PM1 chemical composition have shown that the sulfate contribution may reach more than 50% of aerosol mass, depending on the location. See, for example, Zhang et al., 2007.

2) Page 2, Lines 11-14 (Introduction): a recent and important reference on SO2 sources worldwide, and also on sulfate radiative effects, is Yang et al., 2017.

3) Page 2, Lines 18-20 (Introduction): I recommend that you add a phrase or two to improve the description of the sulfate radiative effects, both direct and indirect. Also, you must include some key references for that.

4) Page 2, Lines 27-29 (Introduction): Do you know of previous studies that promoted integration of data from in situ observations, remote sensing measurements and atmospheric transport modelling? I recommend that you provide an outlook of what has been done before, concerning to data integration from different platforms.

5) Page 2, Line 29 (Introduction): please include a reference for NATALI aerosol typing model.

6) Page 3, Line 15 (Methods): You must give more details about the ground based air quality monitoring site and surroundings. Are there local air pollution sources affecting the site? How is the topography of the surroundings? What are the typical aspects of atmospheric circulation? Are there other air quality monitoring stations nearby?

7) Page 3, Line 15 (Methods): I suggest that you include a description of the general aspects of climate and atmospheric synoptic scale circulation for the study region and season.

8) Page 3, Line 25-28 (Methods): It is very important to include a map showing the location of all stations explored in this manuscript. That will improve understandability

for the readers that are not familiar with EARLINET and with general aspects of Europe geography.

9) Page 5, Line 2 (Methods): I suggest that you briefly explain (2-3 phrases) how a source-receptor model works. What do you need as input? Are there iterations required to tune the model parameters, in order to match model results and observations?

10) Page 5, Line 12-14 (Methods): the term "pure aerosol" usually refers to homogeneous particles made of a single chemical compound. This is not the case of aerosol classes like "continental". Please find another term.

11) Page 5, Lines 28-32 (Methods): it seems that there is a circular reasoning here: you aim to determine aerosol types from Lidar observations (Tables 3 and 6), but, at the same time, you have to assume aerosol types based on NATALI to make use of part of the Lidar observations. Please comment on that.

12) Page 6, Lines 1-2 (Methods): Please include a brief description (1-2 phrases) of the gradient method for detecting aerosol layers, and include more references for that. It is important to state the criteria used to identify an aerosol layer, to provide reproducibility of results. Also, please clarify that you applied the gradient method both for Lidar and CAMS profiles.

13) Page 6, Lines 11-15 (Methods): the analysis of air quality timeseries were performed for how many years of data? How frequent were events like the one you described in the manuscript, in April 2014? What is the objective criteria for "significant excess"?

14) Page 7, Line 14 (Methods): "The release is set to the location of the in situ station". The word "release" is confusing in this context, because it gives the impression that pollutants were set to be released at the in situ station. Maybe "target" would work better here.

15) Page 7, Line 17 (Methods): did you also consider SO2 biogenic sources, like oxidation from DMS? If not, how does it influence your results?

16) Figure 1: SO2 lifetime in the troposphere is typically in the order of hours. Therefore, the SO2 observed at the ground based station may have a contribution of local sources, and possibly cannot be attributed to the regional transport (1-2 days) described in the case study. Please comment on that.

17) Figures 2 and 3: could you convert "model levels" to altitude, to improve understandability of the plots?

18) Page 8, Lines 15-17 (Results): if the diurnal evolution of sulfate and dust layers are correlated with SO2 and PM2.5, and dust is correlated with PM10, can I conclude that all variables are correlated? It would be interesting to see the diurnal evolution of ground based measurements and CAMS. In addition, is this correlation between CAMS and ground based measurements valid for all layers, or just for the lowermost layer?

19) Figure 4: the legend is illegible, text must be enlarged. To better interpret this Figure, it is important to know which profiles correspond to daytime and nighttime (i.e., local time of each plot). Information on the typical planetary boundary layer height at Pillersdorf would also help. It would be interesting to point out whether and when there was an input of aerosols from upper layers to the boundary layer, affecting air quality at Pillersdorf.

20) Figure 5: there are too many lines (altitudes) in the lower plots of sub-figures, it is difficult to interpret. There must be a compromise between completeness and understandability. I suggest that you keep only 3-4 representative altitudes (low, medium, high).

21) Figures 6, 10c, 11c, 12d: you must indicate the locations of the monitoring stations in the maps.

22) Figure 10c: the model calculates SO2 < 1 ug/m3 for layer 1, which is inside the boundary layer, all over Europe. How does it compare to your ground based measurements?

23) Table 3: can you see changes on aerosol properties as they are transported? For example, layers 1 and 2 at Leipzig and Pillersdorf are associated (Table 1). How does aerosol intrinsic and extrinsic properties change along this ~24h transport?

24) Page 9, Lines 19-24: since you did not discuss April 4 in detail, I recommend moving it to the supplementary material, as well as the corresponding figures and tables.

25) Page 10, Lines 10-28: the discussion of trajectories and source regions is rather qualitative. Terms like "medium to smaller contributions" are vague. Could you estimate percent contributions? Also, it is important to recognize limitations and uncertainties of the method.

26) Page 10: meteorological maps for the case study period would help to support the conclusions on aerosol transport. Particularly, trajectories calculated below 2000 m are more prone to uncertainties.

27) Figure 16: it does not contribute significantly to the discussion. I suggest to exclude this figure, or to move it to the supplementary material.

28) Page 11 (conclusion): what are the main advantages of the method you used for this case study, compared to previous database-integration studies? What are the main limitations? How can the method be improved?

Technical corrections:

1) Page 2, Line 17: what do you mean by "key properties"? Optical? Physical? Be more specific.

2) Page 5, Line 11: omit the word "Ref." before citing a reference. That occurs all

through the manuscript, please check.

3) Page 6, Line 6: "The values of the CAMS quantities": please use a more specific term, instead of "quantities".

4) Figure 7: scales are illegible.

5) Figures 7 and 8: could be merged into a single figure. I recommend reformulation of the lidar plots adopting a standard pattern for the contourplots.

6) Figure 9: what are the units for the color map? In addition, the units of longitude should be "degrees", and not "degrees E".

7) Table 3: Please define abbreviations in the table caption, to facilitate interpretation.

References:

Yang, Y., Wang, H., Smith, S. J., Easter, R., Ma, P., Qian, Y., Yu, H., Li, C. and Rasch, P. J.: Global source attribution of sulfate concentration and direct and indirect radiative forcing, Atmos. Chem. Phys., 17, 8903–8922, doi:10.5194/acp-17-8903-2017, 2017.

Zhang, Q., Jimenez, J. L., Canagaratna, M. R., Allan, J. D., Coe, H., Ulbrich, I., Alfarra, M. R., Takami, A., Middlebrook, A. M., Sun, Y. L., Dzepina, K., Dunlea, E., Docherty, K., DeCarlo, P. F., Salcedo, D., Onasch, T., Jayne, J. T., Miyoshi, T., Shimono, A., Hatakeyama, S., Takegawa, N., Kondo, Y., Schneider, J., Drewnick, F., Borrmann, S., Weimer, S., Demerjian, K., Williams, P., Bower, K., Bahreini, R., Cottrell, L., Griffin, R. J., Rautiainen, J., Sun, J. Y., Zhang, Y. M. and Worsnop, D. R.: Ubiquity and dominance of oxygenated species in organic aerosols in anthropogenically-influenced Northern Hemisphere midlatitudes, Geophys. Res. Lett., 34(13), 1–6, doi:10.1029/2007GL029979, 2007.

---

## Author Response (AR1)

Dear referee,

Thank you very much for the comments to our paper.

Here are the answers to your comments. In the following: "RefC" is the comment from Referee, "AuthR" is the author's response and "AuthCM" represents the author's changes to the manuscript. Page and line number refer to the page and line number in the version submitted for discussion.

[Figure]

**Specific Comments**

**Comment 1.**

RefC: "Section 2 (Methodology) is rather lengthy and very detailed. Some parts could perhaps be moved into an Appendix Section."

AuthR: Due to the complexity of the synergy between remote sensing instruments, in situ monitors and modelling, we would prefer to provide the methodology in one section, in the article, even if it is indeed lengthy and very detailed. Splitting it in a short(er) version in the article and details in the Appendix could make it difficult to follow.

AuthCM: none

**Comment 2.**

RefC: "The authors should avoid to mention Trade Names or direct references to companies and commercial instruments, unless absolutely necessary for the understanding of the methods deployed."

AuthR: The trade names of the instruments were deleted from the text. Please note that PollyXT and RALI are the names of the instruments, as used by EARLINET to identify the instruments at the corresponding station, not trade names.

AuthCM: Page 3, Line 21 "a Thermo Scientific Model 43i $SO_2$ Analyzer" was changed to "a $SO_2$ analyzer"

Page 3, Line 22 - Line 23 "Optical Particle Counter GRIMM Dust Monitor Model EDM180," was changed to "optical particle counter"

[Figure]

Page 3, Line 23 - Line 24 "a Thermo Environmental Instruments Ozone Analyzer, model TEI 49C," was changed to "an ozone analyzer,"

Page 4, Line 4 "Jenoptik ceilometers CHM15kx" was changed to "ceilometers"

**Comment 3.**

RefC: "Since the trajectories in Fig.5 indicate sources processes from almost all over Europe (understandable, especially in the lower & mid levels), but also very distant sources (mostly in elevated layers), the authors should show the relevant meteorological maps for the study period (850, 700, 500, 300 & 200 or 250 hPa circulation) to provide physical evidence for 'conflicting' circulations in some of the layers, and especially for the 'outlying regions'. Of course, FLEXTRA ingests the upper air data from ECMWF, but a cross-verification with 'real meteorological data' will make the cases more convincing."

AuthR: There is no reason to not trust the FLEXTRA calculations. However, seeing weather maps may help to understand the prevailing synoptic pattern. Therefore, a few weather maps are now provided in the Supplement.

AuthCM: Added weather maps as supplement.

**Comment 4. Part A**

RefC: "P10 / L27 & 28: - 'No contributions from Europe are seen for these layers.' This may be true for the period in April, as there may not have been any deep convection. However, it would be interesting to also study a summer period with strong convective activity over Central Europe (obviously, in a separate paper !)."

[Figure]

AuthR: The summer periods for the years 2014 – 2017 are under study, and a paper is under preparation.

AuthCM: Page 11, Line 28 Added text: "The spring period studied in this paper is characterized by low, if any, deep convection. For the summer period, one expects however to have a strong convective activity over Central Europe. A study of the summer periods for the years 2014–2017 for the same region was also performed; the results will be presented in a separate paper."

**Comment 4. Part B**

RefC: "P10 / L27 & 28: "I am still a bit skeptical about the long-range transport of pollutants - there would be a significant dilution factor . . .! Unless there are major sources emitting ? An indication of such sources would make your findings more convincing."

AuthR: Flexpart simulates not only the transport due to the large-scale winds but also turbulent diffusion and mixing by subgrid-scale mesoscale motions (A. Stohl et al., 2005). Furthermore, it has implemented the treatment of all loss processes, including dry and wet deposition of gases or aerosols, gravitational settling of particles (S. Eckhardt et al., 2017). Flexpart also has implemented a deep convection scheme. Comprehensive validations of Flexpart were performed for intercontinental air pollution transport, see e.g. [A. Stohl et al Atmos. Environ., 32, 4245–4264, 1998], [A. Stohl and T.Trickl, Geophys. Res., 104, 30,445–30,462, 1999], [N.Kristiansen et al., Geophys. Re. Lett. 42, 588-596, doi 10.1002/2014GL062307, 2015]. Thus, there is no reason to doubt the results.

For the case study presented in this paper, the major sources of $SO_2$ are coal power plants and other industrial facilities (refineries, chemical industry, etc), present in the regions mentioned: Central Europe 'Black triangle', industrialized cities from Morocco, Eastern part of US (e.g. Ohio, New Jersey), Southeastern part of US (e.g. Louisiana,

[Figure]

Alabama). An exhaustive list of US sources is mentioned in the report "U.S. EPA 2014 NEI Version 1.0" [https://www.epa.gov/sites/production/files/2017-04/documents/2014neiv1_profile_final_april182017.pdf] A recent study on $SO_2$ sources worldwide is published in (Y. Yang et al, 2017), which was added to the references.

AuthCM: added references (A. Stohl et al., 2005) to Page 3, Line 4 and (S. Eckhardt et al., 2017) to Page 3, Line 2 for Flexpart and (Y. Yang et al, 2017) to Page 2, after Line 14 for $SO_2$ sources.

**Technical corrections**

**Comment 5.**

RefC: "Figs. 4 & 13 - Key for variables needs to be enlarged P8 / L20 - Height of layers 'amsl' or 'AGL' (also in tables)"

AuthR: The recommended corrections were done in the two figures. AGL was added to text and to the caption for the two figures. No AGL added to the tables, I think it is enough to mention in the text and to add to figure.

AuthCM: Page 3, Line 11: changed to "as ground-level altitudes (AGL)."; added AGL to the caption of the new Figs.4 & 13
* * *
[Figure]

[Figure]

**Fig. 1.** CAMS total aerosol, sulfate and dust profiles for 02 April 2014, Pillersdorf. Grayed area represents the identified sulfate layers. Altitudes are given in km AGL. Local time is UTC+2

[Figure]

[Figure]

**Fig. 2.** CAMS total aerosol, sulfate and dust profiles for 04 April 2014, Pillersdorf. Grayed area represents the identified sulfate layers. Altitudes are given in km AGL. Local time is UTC+2

[Figure]

Atmos. Chem. Phys. Discuss.,
https://doi.org/10.5194/acp-2018-1155-AC2, 2019

[Figure]

Here are the answers to your comments. In the following: "RefC" is the comment from Referee, "AuthR" is the author's response and "AuthCM" represents the author's changes to the manuscript. Page and line number refer to the page and line number in the version submitted for discussion.

[Figure]

**Specific Comments**

**Comment 1.**

RefC: Page 2, Lines 1-2 (Introduction): "worldwide in situ observations of refractory PM1 chemical composition have shown that the sulfate contribution may reach more than 50% of aerosol mass, depending on the location. See, for example, Zhang et al., 2007."

AuthR: This reference was added to the text and it was included in References list.

AuthCM: Page 2, Line 2: Added
"; worldwide in situ observations of refractory $PM_1$ chemical composition have shown that the sulfate contribution may reach more than 50% of aerosol mass, depending on the location (Zhang et al., 2007)."
Added reference (Zhang et al., 2007).

**Comment 2.**

RefC: Page 2, Lines 11-14 (Introduction): "a recent and important reference on SO2 sources worldwide, and also on sulfate radiative effects, is Yang et al., 2017."

AuthR: Added sentence to the text, referencing also the paper.

AuthCM: Added on Page 2, Line 14: "A recent review of $SO_2$ sources worldwide can be found in (Yang et al., 2017)."

[Figure]

**Comment 3.**

RefC: Page 2, Lines 18-20 (Introduction): "I recommend that you add a phrase or two to improve the description of the sulfate radiative effects, both direct and indirect. Also, you must include some key references for that."

AuthR: Done, see changes below.

AuthCM: Page 2, Line 18: added references AEROCOM project, IPCC AR5 for cooling effects of sulfate aerosol
Page 2, Line 20: Added text:
"The direct radiative effects are strongly correlated to the emission sources, while the indirect effects are correlated to both emission sources and cloud cover (Déandreis et al., 2012) (Yang et al., 2017)."

**Comment 4.**

RefC: Page 2, Lines 27-29 (Introduction): "Do you know of previous studies that promoted integration of data from in situ observations, remote sensing measurements and atmospheric transport modelling? I recommend that you provide an outlook of what has been done before, concerning to data integration from different platforms."

AuthR: In the last decade, the synergy of the in situ, remote sensing data and models was used in more atmospheric studies related to long-range transported aerosols and estimation of their potential sources (see for example A. Pappayanis et al. (Sci Total Environ. 2014;500-501:277-94. doi:10.1016/j.scitotenv.2014.08.101, 2014 - C.T. coauthor), D. Nicolae et al. (2013 - C.T. coauthor, Ansmann et al 2018, Eckhardt et al 2008 - P.S. coauthor, Cazacu et al 2012 - C.T. coauthor], [Sauvage et al 2017], [Chalbot et

al 2017],[D.G. Kaskaoutis et al., 2012]).

However, to our best knowledge, there have been no studies combining CAMS-based aerosol data with remote sensing and in situ measurements and transport models.

AuthCM: Added on Page 3, at the end of section "Introduction":
"The synergy of the in situ, remote sensing data and models was used in more atmospheric studies related to long-range transported aerosols and estimation of their potential sources; see for example (Papayannis et al., 2014) for dust, (Nicolae et al., 2013) and (Ansmann et al., 2018) for fires, (Eckhardt et al., 2008) and (Cazacu et al., 2012) for volcanic ash, (Sauvage et al., 2017), (Chalbot et al., 2013) and (Kaskaoutis et al., 2012) for anthropogenic aerosols. However, to our best knowledge, there have been no studies combining CAMS-based aerosol data with remote sensing, in situ measurements and transport models. The assimilation of ground-based remote sensing measurements in CAMS is a long-term goal."

**Comment 5.**

RefC: Page 2, Line 29 (Introduction): "please include a reference for NATALI aerosol typing model."

AuthR: Added reference for NATALI aerosol typing.

AuthCM: Page 2, Line 29 "... and NATALI aerosol-typing model, and atmospheric transport modeling." is replaced by "... and NATALI aerosol-typing model (Nicolae et al., 2018), and atmospheric transport modeling."

[Figure]

**Comment 6.**

RefC: Page 3, Line 15 (Methods): "You must give more details about the ground based air quality monitoring site and surroundings. Are there local air pollution sources affecting the site? How is the topography of the surroundings? What are the typical aspects of atmospheric circulation? Are there other air quality monitoring stations nearby?"

AuthR:

AuthCM: Added text to Page 3, Line 16
"Pillersdorf (315 m) is located in hilly terrain in the northeastern part of Austria, around 60 km north from Vienna. The station is a part of the national background monitoring network and an EMEP background monitoring station. The surroundings are mostly forests and agricultural areas far from strong anthropogenic sources. Austria belongs to the midlatitude climate belt, in the transition between maritime and continental climate, and the weather is dominated mostly by travelling highs and lows. The station provides:"

**Comment 7.**

RefC: Page 3, Line 15 (Methods): "I suggest that you include a description of the general aspects of climate and atmospheric synoptic scale circulation for the study region and season."

AuthR: The description was added to the text for Comment 6.

AuthCM: none

[Figure]

**Comment 8.**

RefC: Page 3, Line 25-28 (Methods): "It is very important to include a map showing the location of all stations explored in this manuscript. That will improve understandability for the readers that are not familiar with EARLINET and with general aspects of Europe geography."

AuthR: OK.

AuthCM: A map has been added to the Supplement.

**Comment 9.**

RefC: Page 5, Line 2 (Methods): "I suggest that you briefly explain (2-3 phrases) how a source-receptor model works. What do you need as input? Are there iterations required to tune the model parameters, in order to match model results and observations?"

AuthR: A more detailed explanation has been introduced. FLEXPART is not tuned or iterated.

AuthCM: Page 5 Line 2: Added after "...or gridded sources"
"(Seibert and Frank, 2004). The model ingests ECMWF 3D meteorological fields and solves the equations for transport, turbulent diffusions and other relevant processes in a Lagrangian framework (Stohl et al., 1998) (Pisso et al., 2019). The sensitivity of a receptor concentration to potential sources is obtained directly as the model output in the case of a backward run (Seibert and Frank, 2004) (Eckhardt et al., 2017)."

[Figure]

**Comment 10.**

RefC: Page 5, Line 12-14 (Methods): "the term "pure aerosol" usually refers to homogeneous particles made of a single chemical compound. This is not the case of aerosol classes like "continental". Please find another term."

AuthR: Changed "pure aerosol" to "typical aerosol".

AuthCM: All occurrences of "pure aerosol" changed to "typical aerosol".

**Comment 11.**

RefC: Page 5, Lines 28-32 (Methods): "it seems that there is a circular reasoning here: you aim to determine aerosol types from Lidar observations (Tables 3 and 6), but, at the same time, you have to assume aerosol types based on NATALI to make use of part of the Lidar observations. Please comment on that."

AuthR: The classes of typical aerosols in NATALI are defined based on the optical properties. If one of the properties is not measured, the type of the aerosol could still be identified based on the other measured properties, which are usually enough to constrain (approximately) the aerosol type. So, on Page 5, Line 28-32, the aerosol type is identified using particle depolarization ratio and AOD at different wavelengths. The lidar ratio is not measured, but it is assumed to be in the range attributed to that class. Please note that the validation of the NATALI model was performed on measurements having all properties measured. So, it is not a circular reasoning, but an estimation based on fully-characterized cases. There is no other way if the measurement is missing. As a cross-check, the type of the aerosol is constrained identifying the potential source with the transport model.

AuthCM: none

[Figure]

**Comment 12.**

RefC: Page 6, Lines 1-2 (Methods): "Please include a brief description (1-2 phrases) of the gradient method for detecting aerosol layers, and include more references for that. It is important to state the criteria used to identify an aerosol layer, to provide reproducibility of results. Also, please clarify that you applied the gradient method both for Lidar and CAMS profiles."

AuthR: Done, see changes to manuscript.

AuthCM: Page 6, Lines 1-2 changed to
"The aerosol layers are identified from the lidar measurements with the gradient method, applied to the RCS profiles (Belegante et al., 2014) (Nicolae et al., 2018). The gradient method is based on the identification of the peaks/valleys from the first derivative applied to the vertical profiles. If two consecutive layers are very close (less than 100 m), these layers are merged into one layer. Also, if the signal to noise ratio in the layer is lower than a threshold (here set to 5), the layer is discarded."
Page 6, Lines 24-26 changed to:
"The layers for the event at the in situ site are then determined by applying the same gradient method as for lidar data processing, but applied to the altitude profiles of aerosol concentrations. The concentrations are computed by multiplying the CAMS mixing ratios and the air density."

**Comment 13a.**

RefC: Page 6, Lines 11-15 (Methods): "the analysis of air quality timeseries were performed for how many years of data? How frequent were events like the one you described in the manuscript, in April 2014?"

[Figure]

AuthR: The analysis of air quality time series were performed for the spring and summer periods for the years 2010–2014. Events as described in the manuscript occur typically 1–2 times per year, between March and April. Unfortunately there were no lidar data to analyze the other events (no trajectories over lidar stations, no high-enough quality data for the corresponding periods). April 2010 was also dominated by the Eyjafjallajökull volcanic eruption.

AuthCM: none

**Comment 13b.**

RefC: Page 6, Lines 11-15 (Methods): "What is the objective criteria for "significant excess"?"

AuthR: The criterion for "significant excess" is 50% above the averaged values for 30 days.

AuthCM: Page 6, Line 13: changed "...is identified, " to "...is identified (values exceed by 50% the averaged values for 30 days),"

**Comment 14.**

RefC: Page 7, Line 14 (Methods): ""The release is set to the location of the in situ station". The word "release" is confusing in this context, because it gives the impression that pollutants were set to be released at the in situ station. Maybe "target" would work better here.""

AuthR: "release" was changed with "receptor".

AuthCM: "The receptor is set to the location"

[Figure]

**Comment 15.**

RefC: Page 7, Line 17 (Methods): "did you also consider SO2 biogenic sources, like oxidation from DMS? If not, how does it influence your results?"

AuthR: yes, the Flexpart model considers all the sulfate sources, including $SO_2$ biogenic sources. Also, for the trajectories at high altitude over ocean, the influence of biogenic sources is small.

AuthCM: none

**Comment 16.**

RefC: Figure 1: "SO2 lifetime in the troposphere is typically in the order of hours. Therefore, the SO2 observed at the ground based station may have a contribution of local sources, and possibly cannot be attributed to the regional transport (1-2 days) de-scribed in the case study. Please comment on that."

AuthR: As we mentioned before (see answer to comment 6), the station Pillersdorf is a regional background site, with no significant local anthropogenic sources, especially of $SO_2$ (by the requirements of EMEP "background site" and also because the $SO_2$ emissions are generally very low in Austria). The $SO_2$ measured at Pillersdorf is the result of the sulfate transported together with dust.

AuthCM: none

[Figure]

**Comment 17.**

RefC: Figures 2 and 3: "could you convert "model levels" to altitude, to improve understandability of the plots?"

AuthR: we would prefer to keep the model levels in these plots because the value (as altitude) of each model level is variable, it depends on the meteorological conditions (temperature, humidity, etc) that are variable in time. For each case, the altitude can be computed using the geopotential heights.

AuthCM: none

**Comment 18.**

RefC: Page 8, Lines 15-17 (Results): "if the diurnal evolution of sulfate and dust layers are correlated with SO2 and PM2.5, and dust is correlated with PM10, can I conclude that all variables are correlated? It would be interesting to see the diurnal evolution of ground based measurements and CAMS. In addition, is this correlation between CAMS and ground based measurements valid for all layers, or just for the lowermost layer?"

AuthR: There is a correlation between sulfate and dust layers and values of $SO_2$, $PM_{2.5}$ and $PM_{10}$ from in-situ measurements, but it is important to analyze each case separately to find the correlation factor between them. Also, the correlation between CAMS and ground based measurements is valid for the lowermost layer (in situ data are assimilated in CAMS). For the rest of the layers, not much can be said as CAMS does not assimilate yet ground-based lidar data.

AuthCM: none

[Figure]

**Comment 19.**

RefC: Figure 4: "the legend is illegible, text must be enlarged. To better interpret this Figure, it is important to know which profiles correspond to daytime and nighttime (i.e., local time of each plot). Information on the typical planetary boundary layer height at Pillersdorf would also help. It would be interesting to point out whether and when there was an input of aerosols from upper layers to the boundary layer, affecting air quality at Pillersdorf."

AuthR: We have improved the figure and added a sentence on boundary-layer heights. We also added in the caption of Fig. 4 the UTC–local time difference.

AuthCM: Changed fonts in the legend of Fig. 4; modified caption: changed "aerosol" to "total aerosol"; added "Local time is UTC+2.".

Added on Page 6, Line 31:
"During the period under investigation, with low wind speeds and mostly clear skies, the boundary-layer height varied at Pillersdorf from less than 100 m at night to about 1500 m in the afternoon."

**Comment 20.**

RefC: Figure 5: "there are too many lines (altitudes) in the lower plots of sub-figures, it is difficult to interpret. There must be a compromise between completeness and understandability. I suggest that you keep only 3-4 representative altitudes (low, medium, high)."

AuthR: The figure was changed, keeping only the trajectories passing over the lidar stations involved in this analysis.

[Figure]

AuthCM: New Fig. 5

**Comment 21.**

RefC: "Figures 6, 10c, 11c, 12d: you must indicate the locations of the monitoring stations in the maps."

AuthR: The figures were changed adding the location of the monitoring stations.

AuthCM: New figures 6, 10c, 11c, 12d

**Comment 22.**

RefC: Figure 10c: "the model calculates $SO_2 < 1$ ug/m3 for layer 1, which is inside the boundary layer, all over Europe. How does it compare to your ground based measurements?"

AuthR: There was a bug in superposing the concentration distribution on map for the zoomed distribution (only). The figures 10c, 11c and 12d were replaced with figures with correct data.

There is a good agreement between model and ground based measurements, as can be seen from Fig. 1 and Fig. 10c.

AuthCM: the figures 10c, 11c and 12c were replaced.

[Figure]

**Comment 23.**

RefC: Table 3: "can you see changes on aerosol properties as they are transported? For example, layers 1 and 2 at Leipzig and Pillersdorf are associated (Table 1). How does aerosol intrinsic and extrinsic properties change along this $\approx$ 24h transport?"

AuthR: Yes, changes on aerosol properties can be observed for the correlated layers. For layer 1, the aerosol transported from Leipzig to Pillersdorf is mixed with the aerosols accumulated inside the PBL (the trajectory is under 1500 m), leading to an increase of the sulfate fraction in aerosols. The aerosol size increase also. These are reflected by the decrease in the lidar ratio and AE, measured at Leipzig and computed at Pillersdorf.

For Layer 2 (above PBL), the depolarization ratio, sulfate fraction and AE decreases, due to chemical processes (aging) and aerosol removal processes. The lidar ratio slightly increases, due to the decrease of the sulfate fraction from aerosol.

AuthCM: none

**Comment 24.**

RefC: Page 9, Lines 19-24: "since you did not discuss April 4 in detail, I recommend moving it to the supplementary material, as well as the corresponding figures and tables."

AuthR: We would prefer to keep the event in the article, as it is consistent with the event from Apr 02 and emphasize the main message of the paper (long-range transport of sulfate aerosols must be considered for local changes, as it can have non-negligible effects).

AuthCM: none

[Figure]

**Comment 25.**

RefC: Page 10, Lines 10-28: "the discussion of trajectories and source regions is rather qualitative. Terms like "medium to smaller contributions" are vague. Could you estimate percent contributions? Also, it is important to recognize limitations and uncertainties of the method."

AuthR: I could compute the contributions of the sources, but they would be obtained from the transport model, they would be a rough estimation, therefore I prefer to give only qualitative values. For confident quantitative results, I would need more measured data, time-dependent, not only monthly averaged sources, to be able to compute the central value and to compute the uncertainties.

AuthCM: none

**Comment 26.**

RefC: Page 10: "meteorological maps for the case study period would help to support the conclusions on aerosol transport. Particularly, trajectories calculated below 2000 m are more prone to uncertainties."

AuthR: The meteorological maps will be added to the Supplement of the paper.

AuthCM: meteorological maps in the supplement.

**Comment 27.**

RefC: Figure 16: "it does not contribute significantly to the discussion. I suggest to exclude this figure, or to move it to the supplementary material."

[Figure]

AuthR: OK

AuthCM: Fig. 16 was moved to the Supplement of the paper.

**Comment 28.**

RefC: Page 11 (conclusion): "what are the main advantages of the method you used for this case study, compared to previous database-integration studies? What are the main limitations? How can the method be improved?"

AuthR: I think this is covered by the text added in Introduction as answer to Comment 4 and by the text from Page 11, lines 25 - 31.

AuthCM: none

**Technical corrections**

**Comment 29.**

RefC: Page 2, Line 17: "what do you mean by "key properties"? Optical? Physical? Be more specific."

AuthR: "key properties" means "optical, physical and chemical properties".

AuthCM: Page 2, Line 17
"The key properties" was changed to "The optical, physical and chemical properties"

[Figure]

**Comment 30.**

RefC: Page 5, Line 11: "omit the word "Ref." before citing a reference. That occurs all through the manuscript, please check."

AuthR: OK.

AuthCM: removed "Ref. "

**Comment 31.**

RefC: Page 6, Line 6: ""The values of the CAMS quantities": please use a more specific term, instead of "quantities"."

AuthR: "the CAMS quantities" were specified in text at page 4, line 22 - line 28.

AuthCM: Page 6, Line 6: "... the CAMS quantities" was changed to "... the CAMS products (mixing ratios, temperature, specific humidity, etc)"

**Comment 32.**

RefC: Figure 7: "scales are illegible."

AuthCM: Figure 7 The fonts for scales were increased to be more legible.

[Figure]

**Comment 33.**

RefC: Figures 7 and 8: "could be merged into a single figure. I recommend reformulation of the lidar plots adopting a standard pattern for the contourplots."

AuthR: The lidar plots are taken in the format available in the Earlinet database. I prefer to keep the format used by the Earlinet database for consistency.

AuthCM: none

**Comment 34.**

RefC: Figure 9: "what are the units for the color map? In addition, the units of longitude should be "degrees", and not "degrees E"."

AuthR: The units for color map are seconds. The units of longitude and latitude were corrected.

AuthCM: the figures 9, 10, 11, 12 and 15 were replaced with the figures with units.

**Comment 35.**

RefC: Table 3: "Please define abbreviations in the table caption, to facilitate interpretation."

AuthR: The abbreviations are defined in the text.

AuthCM: none

[Figure]

[Figure]

**Fig. 1.** CAMS total aerosol, sulfate and dust profiles for 02 April 2014, Pillersdorf. Grayed area represents the identified sulfate layers. Altitudes are given in km AGL. Local time is UTC+2.

[Figure]

**Fig. 2.** Pattern of back-trajectories (upper plot of sub-figure) and their altitude profile, including overpassed lidar stations (lower plot of sub-figure) for Pillersdorf, 02 April 2014.

[Figure]

**Fig. 3.** Pattern of forward-trajectories (upper plot) and their altitude profile, including over-passed lidar stations (lower plot) for Pillersdorf, 02 April 2014, 06:00.

[Figure]

[Figure]

(a) Munich, 01 April 2014 Ceilometer YALIS

[Figure]

(b) Garmisch, 01 April 2014 Ceilometer

**Fig. 4.** Logarithm of the range corrected signal at 1064 nm, 24 h, for Munich (a) and Garmisch (b) stations. The red line boxes represent the identified layers.

[Figure]

[Figure]

(a) Layer L1

(b) Layer L2

(c) Layer L3

(d) Total column

**Fig. 5.** Source-receptor sensitivity for layer L1 (a), L2 (b) and L3 (c) and total column (d), Pillersdorf, 02 April, 6:00.

[Figure]

[Figure]

(a) Pillersdorf, 02 April, 06:00, layer L1

(b) Leipzig, 31 March, 18:00, layer corresponding to L1

[Figure]

(c) Pillersdorf, 02 April, 06:00, layer L1, zoomed

**Fig. 6.** Relative distributions of SO2 sources for Pillersdorf layer L1 (a), Leipzig (b); zoomed distribution for Pillersdorf layer L1 (c).

[Figure]

[Figure]

(a) Pillersdorf, 02 April, 06:00, layer L2

(b) Leipzig, 31 March, 23:00, layer corresponding to L2

[Figure]

(c) Pillersdorf, 02 April, 06:00, layer L2, zoomed

**Fig. 7.** Relative distributions of SO2 sources for Pillersdorf layer L2 (a), Leipzig (b); zoomed distribution for Pillersdorf layer L2 (c).

[Figure]

[Figure]

(a) Pillersdorf, 02 April, 06:00, layer L3

(b) Munich, 01 April, 05:00, layer corresponding to L3

(c) Bucharest, 03 April, 13:00, layer corresponding to L3

(d) Pillersdorf, 02 April, 06:00, layer L3, zoomed

**Fig. 8.** Relative distributions of SO2 sources for Pillersdorf layer L3 (a), Munich (b), Bucharest (c); zoomed distribution for Pillersdorf layer L3 (d).

[Figure]

[Figure]

(a) Pillersdorf, 04 April, 12:00, layer L1

[Figure]

(b) Pillersdorf, 04 April, 12:00, layer L2

**Fig. 9.** Source-receptor sensitivity for layer L1 (a) and L2 (b), Pillersdorf, 04 April, 12:00

[Figure]

Atmos. Chem. Phys. Discuss.,
https://doi.org/10.5194/acp-2018-1155-AC3, 2019

[Figure]

Here are some technical corrections from the authors. Comment 1 to Comment 10 are minor language corrections. For maps in the Figures 5, 6, 9, 10, 11, 12 and 15, the color of the continents was changed from etopo color style to gray style (continents) and white (oceans) to make the maps more clear (Comment 11). Also included are the changes due to moving the GIOVANNI maps to the Supplement. In the following "AuthCM" represents the author's changes to the manuscript. Page and line number refer to the page and line number in the version submitted for discussion.

[Figure]

**Technical corrections:**

**Comment 1.**

AuthCM:
Page 2, Line 21
Changed in text: "The study was based on" to
"The study is based on"

Page 2, Line 22 - Line 25
Rephrased text to
"to assess the relation between the excess with respect to monthly averaged values observed in the in situ measurements of $SO_2$, $O_3$, $PM_{2.5}$ and $PM_{10}$ at the Austrian air quality background station Pillersdorf at the beginning of Apr 2014 with aerosols layers observed in lidar measurements at the closest EARLINET stations around Austria and with tropospheric sulfate aerosols as found in Copernicus Atmosphere Monitoring Service (CAMS) products (CAMS, 2018)"

Page 2, Line 27
Changed in text: "to estimate the sulfate aerosols potential sources." to
"to estimate the potential sources of sulfate aerosols"

**Comment 2.**

AuthCM:
Page 3, Line 10
Changed in text: "in the format HH:mm, H being the hour and m the minutes" to
"in the format HH:mm, HH being the hour and mm the minutes"

[Figure]

Page 3, Line 13
Added: "In the plots, the stations are represented as: Pillersdorf (red circle), Leipzig (green circle), Munich (magenta triangle), Garmisch (blue rhombus), Bucharest (black square)."

Page 3, Line 17
Changed in text: "daily mean concentration and the maximum value per day of half-an hour averaged concentrations for" to
"daily mean concentration and the maximum half-hour mean value per day for"

Page 3, Line 19
"averaged" was changed with "mean"

**Comment 3.**

AuthCM:
Page 4, Line 30
Changed in text: "FLEXPART and FLEXTRA models were used in this paper for atmospheric transport modelling." to
"In this paper, the models FLEXPART and FLEXTRA were used for atmospheric transport modelling."

**Comment 4.**

AuthCM:
Page 5, Line 3 - Line 4
Changed in text: "parcels by mean winds, ignoring turbulence and convection, and do

not represent" to
"parcels by mean winds, ignoring turbulence and convection, and do not provide"

Page 5, Line 6
Changed "x" with "×"

**Comment 5.**

AuthCM:
Page 8, Line 4
Changed in text: ""total aerosols" (sum of all species defined in CAMS data) (a), for sulfate (b) and for dust (c)." to
""total aerosols" (sum of all species defined in CAMS data), for sulfate and for dust."

Page 8, Line 8
Changed in text: "for Munich (a), Leipzig (b) and Bucharest (c)." to
"for Munich, Leipzig and Bucharest."

Page 8, Line 21
Changed in text: "Fig. 5 for 00:00 (a), 06:00 (b), 12:00 (c) and 18:00 (d)." to
"Fig. 5 for 00:00, 06:00, 12:00 and 18:00."

**Comment 6.**

AuthCM:
Page 9, Line 2 - Line 3
Changed in text: "for the layers L1 (a), L2 (b), L3 (c) and total column (d)" to
"for the layers L1, L2, L3 and total column"

[Figure]

Page 9, Line 34
Changed in text: "(not shown in this paper)" to
"(not shown)"

**Comment 7.**

AuthCM:
Page 10, Line 11 - Line 17
Changed in text "transverse" to
"traverse"

Page 10, Line 19
Changed in text "at Pillersdorf: Southern and" to
"at Pillersdorf, namely Southern and"

**Comment 8.**

AuthCM:
Page 11, Line 7
Changed in text "Garmisch-Partenkirchen, Pillersdorf and Bucharest" to
"Garmisch-Partenkirchen, Pillersdorf and Bucharest, and"

Page 11, Line 8
Changed in text: "the adsorption of the $SO_2$ on the dust mineral oxides compounds." to
"the adsorption of the $SO_2$ on oxides contained in the mineral dust."

[Figure]

**Comment 9.**

AuthCM:
Page 22, Figure 7
Changed in caption: "log(range corrected signal)" to
"Logarithm of the range corrected signal"

**Comment 10.**

AuthCM:
Changed in text: "Apr" to "April" and "Mar" to "March"

**Comment 11.**

AuthCM:
Changed color of maps to gray (continents) and white (oceans) in the Figures 5, 6, 9, 10, 11, 12 and 15.

**Technical corrections: content moved to Supplement**

AuthCM:
Page 11, Line 10 - Line 13: moved to Supplement.

Changes in text, in the Supplement: "Monthly-averaged maps of column mass density for sulfate are available from EarthData NASA GIOVANNI (NASA,2018) online data system" to

"Time averaged maps of sulfate column mass density, monthly, are available from EarthData NASA GIOVANNI online data system (NASA,2018)."

Page 12, Line 8 - Line 10: acknowledgement to GIOVANNI moved to Supplement.

Moved Fig. 16 to Supplement.

[Figure]

**Analysis of Sulfate Aerosols over Austria: A Case Study**

Camelia Talianu[1,2] and Petra Seibert[1]

[1]Institute of Meteorology, University of Natural Resources and Life Sciences, Vienna, Austria
[2]National Institute of R&D for Optoelectronics, Magurele, Romania

**Correspondence:** Camelia Talianu (camelia.talianu@boku.ac.at)

Dear editor,

Here are the changes between the version submitted for discussion and the revised version, updated as result of the referee comments and author supplementary comments. No other changes were done.

In the following: for "Changes in the paper", the page and line number refer to the page and line number in the latexdiff version; for "Paper supplement" (text added in the Supplement, added in the revised version) the page and line number refer to the page and line number in the supplement file.

The reason of each change is given in the format "ACx: Comment xx: AuthCM: ccc" where

– AC1: 'answer_acp-2018-1155-RC1', Camelia Talianu, 25 Mar 2019 (link: answer to comments from Referee 1)

– AC2: 'answer_acp-2018-1155-RC2', Camelia Talianu, 25 Mar 2019 (link: answer to comments from Referee 2)

– AC3: 'Author comments: acp-2018-1155', Camelia Talianu, 25 Mar 2019 (link: author comments)

The RefC1 (RefC2) is a copy of RefC for referee 1 (2) from AC1 (AC2); AuthR and AuthCM are the corresponding author response and author changes in the manuscripts, also copies from AC1, AC2, AC3, respectively.

**Changes in the paper**

Complete list of changes for the paper, as shown in the latexdiff file.

1. Page 1, Line 1: change resulted from AC3: Comment 10.
   AuthCM: Changed in text: "Apr" to "April"

2. Page 2, Lines 2–3: change resulted from AC2: Comment 1.
   RefC2: Page 2, Lines 1–2 (Introduction): "worldwide in situ observations of refractory PM1 chemical composition have shown that the sulfate contribution may reach more than 50% of aerosol mass, depending on the location. See, for example, Zhang et al., 2007."
   AuthR: This reference was added to the text and it was included in References list.
   AuthCM: Added "; worldwide in situ observations of refractory $PM_1$ chemical composition have shown that the sulfate contribution may reach more than 50% of aerosol mass, depending on the location (Zhang et al., 2007)."
   Added reference (Zhang et al., 2007).

3. Page 2, Line 17: change resulted from AC2: Comment 2.

   RefC2: Page 2, Lines 11–14 (Introduction): "a recent and important reference on SO2 sources worldwide, and also on sulfate radiative effects, is Yang et al., 2017."

   AuthR: Added sentence to the text, referencing also the paper.

   AuthCM: "A recent review of $SO_2$ sources worldwide can be found in (Yang et al., 2017)."

4. Page 2, Line 20: change resulted from AC2: Comment 29.

   RefC2: Page 2, Line 17: "what do you mean by "key properties"? Optical? Physical? Be more specific."

   AuthR: "key properties" means "optical and chemical properties".

   AuthCM: "The key properties" was changed to "The optical, physical and chemical properties"

5. Page 2, Lines 22–25: change resulted from AC2: Comment 3.

   RefC2: Page 2, Lines 18–20 (Introduction): "I recommend that you add a phrase or two to improve the description of the sulfate radiative effects, both direct and indirect. Also, you must include some key references for that."

   AuthCM: added references AEROCOM project, IPCC AR5 for cooling effects of sulfate aerosol and added text: "The direct radiative effects are strongly correlated to the emission sources, while the indirect effects are correlated to both emission sources and cloud cover (Déandreis et al., 2012) (Yang et al., 2017)."

6. Page 2, Line 29: change resulted from AC3: Comment 10.

   AuthCM: Changed in text: "Apr" to "April"

7. Page 2, Lines 30–32: change resulted from AC3: Comment 1.

   AuthCM: Rephrased text: "to assess the relation between the excess with respect to monthly averaged values observed in the in situ measurements of $SO_2$, $O_3$, $PM_{2.5}$ and $PM_{10}$ at the Austrian air quality background station Pillersdorf at the beginning of Apr 2014 with aerosols layers observed in lidar measurements at the closest EARLINET stations around Austria and with tropospheric sulfate aerosols as found in Copernicus Atmosphere Monitoring Service (CAMS) products (CAMS, 2018)"

8. Page 2, Line 33: change resulted from AC3: Comment 1.

   AuthCM: Changed in text: "to estimate the sulfate aerosols potential sources." to "to estimate the potential sources of sulfate aerosols"

9. Page 3, Line 1: change resulted from AC3: Comment 1.

   AuthCM: Changed in text: "The study was based on" to "The study is based on"

10. Page 3, Line 3: change resulted from AC2: Comment 5.

    RefC2: Page 2, Line 29 (Introduction): "please include a reference for NATALI aerosol typing model."

    AuthR: Added reference for NATALI aerosol typing.

AuthCM: "... and NATALI aerosol-typing model, and atmospheric transport modeling." is replaced by "... and NATALI aerosol-typing model (Nicolae et al., 2018), and atmospheric transport modeling."

11. Page 3, Lines 9–12: change resulted from AC1: Comment 4. Part B.

RefC1: "P10 / L27 & 28: "I am still a bit skeptical about the long-range transport of pollutants - there would be a significant dilution factor . . .! Unless there are major sources emitting ? An indication of such sources would make your findings more convincing."

AuthR: Flexpart simulates not only the transport due to the large-scale winds but also turbulent diffusion and mixing by subgrid-scale mesoscale motions (A. Stohl et al., 2005). Furthermore, it has implemented the treatment of all loss processes, including dry and wet deposition of gases or aerosols, gravitational settling of particles (S. Eckhardt et al., 2017). Flexpart also has implemented a deep convection scheme. Comprehensive validations of Flexpart were performed for intercontinental air pollution transport, see e.g. [A. Stohl et al Atmos. Environ., 32, 4245–4264, 1998], [A. Stohl and T.Trickl, Geophys. Res., 104, 30,445–30,462, 1999], [N.Kristiansen et al., Geophys. Re. Lett. 42, 588-596, doi 10.1002/2014GL062307, 2015]. Thus, there is no reason to doubt the results.

For the case study presented in this paper, the major sources of $SO_2$ are coal power plants and other industrial facilities (refineries, chemical industry, etc), present in the regions mentioned: Central Europe 'Black triangle', industrialized cities from Morocco, Eastern part of US (e.g. Ohio, New Jersey), Southeastern part of US (e.g. Louisiana, Alabama). An exhaustive list of US sources is mentioned in the report "U.S. EPA 2014 NEI Version 1.0" [https://www.epa.gov/sites/production/files/2017-04/documents/2014neiv1_profile_final_april182017.pdf] A recent study on $SO_2$ sources worldwide is published in (Y. Yang et al, 2017), which was added to the references.

AuthCM: added references (S. Eckhardt et al., 2017) and (A. Stohl et al., 2005) for Flexpart

12. Page 3, Lines 14–19: change resulted from AC2: Comment 4.

RefC2: Page 2, Lines 27–29 (Introduction): "Do you know of previous studies that promoted integration of data from in situ observations, remote sensing measurements and atmospheric transport modelling? I recommend that you provide an outlook of what has been done before, concerning to data integration from different platforms."

AuthR: In the last decade, the synergy of the in situ, remote sensing data and models was used in more atmospheric studies related to long-range transported aerosols and estimation of their potential sources (see for example A. Pappayanis et al. (Sci Total Environ. 2014;500-501:277-94. doi:10.1016/j.scitotenv.2014.08.101, 2014 - C.T. coauthor), D. Nicolae et al. (2013 - C.T. coauthor, Ansmann et al 2018, Eckhardt et al 2008 - P.S. coauthor, Cazacu et al 2012 - C.T. coauthor], [Sauvage et al 2017], [Chalbot et al 2017],[D.G. Kaskaoutis et al., 2012]).

However, to our best knowledge, there have been no studies combining CAMS-based aerosol data with remote sensing and in situ measurements and transport models.

AuthCM: Added: "The synergy of the in situ, remote sensing data and models was used in more atmospheric studies related to long-range transported aerosols and estimation of their potential sources; see for example (Papayannis et al., 2014) for dust, (Nicolae et al., 2013) and (Ansmann et al., 2018) for fires, (Eckhardt et al., 2008) and (Cazacu et al.,

2012) for volcanic ash, (Sauvage et al., 2017), (Chalbot et al., 2013) and (Kaskaoutis et al., 2012) for anthropogenic aerosols. However, to our best knowledge, there have been no studies combining CAMS-based aerosol data with remote sensing, in situ measurements and transport models. The assimilation of ground-based remote sensing measurements in CAMS is a long-term goal."

13. Page 3, Line 23: change resulted from AC3: Comment 2.

    AuthCM: Changed in text: "in the format HH:mm, H being the hour and m the minutes" to
    "in the format HH:mm, HH being the hour and mm the minutes"

14. Page 3, Line 24: change resulted from AC1: Comment 5.

    RefC1: "Figs. 4 & 13 - Key for variables needs to be enlarged P8 / L20 - Height of layers 'amsl' or 'AGL' (also in tables)"

    AuthR: The recommended corrections were done in the two figures. AGL was added to text and to the caption for the two figures. No AGL added to the tables, I think it is enough to mention in the text and to add to figure.

    AuthCM: added "(AGL)."

15. Page 3, Lines 26–27: change resulted from AC3: Comment 2.

    AuthCM: Added: "In the plots, the stations are represented as: Pillersdorf (red circle), Leipzig (green circle), Munich (magenta triangle), Garmisch (blue rhombus), Bucharest (black square)."

16. Page 3, Lines 30–32, Page 4, Lines 1–2: change resulted from AC2: Comment 6.

    RefC2: Page 3, Line 15 (Methods): "You must give more details about the ground based air quality monitoring site and surroundings. Are there local air pollution sources affecting the site? How is the topography of the surroundings? What are the typical aspects of atmospheric circulation? Are there other air quality monitoring stations nearby?"

    AuthR:

    AuthCM: Added: "Pillersdorf (315 m) is located in hilly terrain in the northeastern part of Austria, around 60 km north from Vienna. The station is a part of the national background monitoring network and an EMEP background monitoring station. The surroundings are mostly forests and agricultural areas far from strong anthropogenic sources. Austria belongs to the midlatitude climate belt, in the transition between maritime and continental climate, and the weather is dominated mostly by travelling highs and lows. The station provides:"

17. Page 4, Line 3: change resulted from AC3: Comment 2.

    AuthCM: Changed in text: "daily mean concentration and the maximum value per day of half-an hour averaged concentrations for" to
    "daily mean concentration and the maximum half-hour mean value per day for"

18. Page 4, Line 6: change resulted from AC3: Comment 2.

    AuthCM: "averaged" was changed with "mean"

19. Page 4, Lines 8–11: change resulted from AC1: Comment 2.

   RefC1: "The authors should avoid to mention Trade Names or direct references to companies and commercial instruments, unless absolutely necessary for the understanding of the methods deployed."

   AuthR: The trade names of the instruments were deleted from the text. Please note that PollyXT and RALI are the names of the instruments, as used by EARLINET to identify the instruments at the corresponding station, not trade names.

   AuthCM: "a Thermo Scientific Model 43i $SO_2$ Analyzer" was changed to "a $SO_2$ analyzer"

   "Optical Particle Counter GRIMM Dust Monitor Model EDM180," was changed to "optical particle counter"

   "a Thermo Environmental Instruments Ozone Analyzer, model TEI 49C," was changed to "an ozone analyzer,"

20. Page 4, Line 22: change resulted from AC1: Comment 2.

   RefC1: "The authors should avoid to mention Trade Names or direct references to companies and commercial instruments, unless absolutely necessary for the understanding of the methods deployed."

   AuthCM: "Jenoptik ceilometers CHM15kx" was changed to "ceilometers"

21. Page 5, Line 16: change resulted from AC3: Comment 3.

   AuthCM: Changed in text: "FLEXPART and FLEXTRA models were used in this paper for atmospheric transport modelling." to

   "In this paper, the models FLEXPART and FLEXTRA were used for atmospheric transport modelling."

22. Page 5, Lines 23–26: change resulted from AC2: Comment 9.

   RefC2: Page 5, Line 2 (Methods): "I suggest that you briefly explain (2-3 phrases) how a source-receptor model works. What do you need as input? Are there iterations required to tune the model parameters, in order to match model results and observations?"

   AuthR: A more detailed explanation has been introduced. FLEXPART is not tuned or iterated.

   AuthCM: Added after "...or gridded sources"

   "(Seibert and Frank, 2004). The model ingests ECMWF 3D meteorological fields and solves the equations for transport, turbulent diffusions and other relevant processes in a Lagrangian framework (Stohl et al., 1998) (Pisso et al., 2019). The sensitivity of a receptor concentration to potential sources is obtained directly as the model output in the case of a backward run (Seibert and Frank, 2004) (Eckhardt et al., 2017)."

23. Page 5, Line 28: change resulted from AC3: Comment 4.

   AuthCM: Changed in text: "parcels by mean winds, ignoring turbulence and convection, and do not represent" to

   "parcels by mean winds, ignoring turbulence and convection, and do not provide"

24. Page 5, Line 30: change resulted from AC3: Comment 4.

   AuthCM: Changed "x" with "×"

25. Page 6, Line 2: change resulted from AC2: Comment 30.

   RefC2: Page 5, Line 11: "omit the word "Ref." before citing a reference. That occurs all through the manuscript, please

check."

AuthR: OK.

AuthCM: removed "Ref. "

26. Page 6, Lines 3–5: change resulted from AC2: Comment 10.

RefC2: Page 5, Line 12-14 (Methods): "the term "pure aerosol" usually refers to homogeneous particles made of a single chemical compound. This is not the case of aerosol classes like "continental". Please find another term."

AuthR: Changed "pure aerosol" to "typical aerosol".

AuthCM: All occurrences of "pure aerosol" changed to "typical aerosol"; added in text "(called "pure aerosol" in the reference)".

27. Page 6, Lines 25–29: change resulted from AC2: Comment 12.

RefC2: Page 6, Lines 1–2 (Methods): "Please include a brief description (1-2 phrases) of the gradient method for detecting aerosol layers, and include more references for that. It is important to state the criteria used to identify an aerosol layer, to provide reproducibility of results."

AuthR: Done, see changes to manuscript.

AuthCM: changed to "The aerosol layers are identified from the lidar measurements with the gradient method, applied to the RCS profiles (Belegante et al., 2014) (Nicolae et al., 2018). The gradient method is based on the identification of the peaks/valleys from the first derivative applied to the vertical profiles. If two consecutive layers are very close (less than 100 m), these layers are merged into one layer. Also, if the signal to noise ratio in the layer is lower than a threshold (here set to 5), the layer is discarded."

28. Page 6, Line 30: change resulted from AC2: Comment 30.

RefC2: Page 5, Line 11: "omit the word "Ref." before citing a reference. That occurs all through the manuscript, please check."

AuthR: OK.

AuthCM: removed "Ref. "

29. Page 7, Line 2: change resulted from AC2: Comment 31.

RefC2: Page 6, Line 6: ""The values of the CAMS quantities": please use a more specific term, instead of "quantities"."

AuthR: "the CAMS quantities" were specified in text at page 4, line 22 - line 28.

AuthCM: "... the CAMS quantities" was changed to "... the CAMS products (mixing ratios, temperature, specific humidity, etc)"

30. Page 7, Line 9: change resulted from AC2: Comment 13b.

RefC2: Page 6, Lines 11–15 (Methods): "What is the objective criteria for "significant excess"?"

AuthR: The criterion for "significant excess" is 50% above the averaged values for 30 days.

AuthCM: changed "...is identified, " to "...is identified (values exceed by 50% the averaged values for 30 days),"

31. Page 7, Lines 11–12: change resulted from AC3: Comment 10.

AuthCM: Changed in text: "Apr" to "April" and "Mar" to "March"

32. Page 7, Lines 21–23: change resulted from AC2: Comment 12.

RefC2: Page 6, Lines 1–2 (Methods): " Also, please clarify that you applied the gradient method both for Lidar and CAMS profiles."

AuthR: Done, see changes to manuscript.

AuthCM: changed to: "The layers for the event at the in situ site are then determined by applying the same gradient method as for lidar data processing, but applied to the altitude profiles of aerosol concentrations. The concentrations are computed by multiplying the CAMS mixing ratios and the air density."

33. Page 7, Lines 28–29: change resulted from AC2: Comment 19.

RefC2: Figure 4: "Information on the typical planetary boundary layer height at Pillersdorf would also help. It would be interesting to point out whether and when there was an input of aerosols from upper layers to the boundary layer, affecting air quality at Pillersdorf."

AuthR: We added a sentence on boundary-layer heights.

AuthCM: Added: "During the period under investigation, with low wind speeds and mostly clear skies, the boundary-layer height varied at Pillersdorf from less than 100 m at night to about 1500 m in the afternoon."

34. Page 8, Line 5: change resulted from AC2: Comment 30.

RefC2: Page 5, Line 11: "omit the word "Ref." before citing a reference. That occurs all through the manuscript, please check."

AuthR: OK.

AuthCM: removed "Ref. "

35. Page 8, Line 11: change resulted from AC2: Comment 14.

RefC2: Page 7, Line 14 (Methods): ""The release is set to the location of the in situ station". The word "release" is confusing in this context, because it gives the impression that pollutants were set to be released at the in situ station. Maybe "target" would work better here.""

AuthR: "release" was changed with "receptor".

AuthCM: "The receptor is set to the location"

36. Page 8, Lines 27–29: change resulted from AC3: Comment 10.

AuthCM: Changed in text: "Apr" to "April" and "Mar" to "March"

37. Page 8, Line 33: change resulted from AC3: Comment 5.

AuthCM: Changed in text: ""total aerosols" (sum of all species defined in CAMS data) (a), for sulfate (b) and for dust (c)." to

""total aerosols" (sum of all species defined in CAMS data), for sulfate and for dust."

38. Page 9, Line 1: change resulted from AC3: Comment 10.

    AuthCM: Changed in text: "Apr" to "April"

39. Page 9, Line 4: change resulted from AC3: Comment 5.

    AuthCM: Changed in text: "for Munich (a), Leipzig (b) and Bucharest (c)." to

    "for Munich, Leipzig and Bucharest."

40. Page 9, Line 6: change resulted from AC3: Comment 10.

    AuthCM: Changed in text: "Apr" to "April"

41. Page 9, Line 8: change resulted from AC3: Comment 10.

    AuthCM: Changed in text: "Apr" to "April"

42. Page 9, Line 17: change resulted from AC3: Comment 10.

    AuthCM: Changed in text: "Apr" to "April"

43. Page 9, Line 17: change resulted from AC3: Comment 5.

    AuthCM: Changed in text: "Fig. 5 for 00:00 (a), 06:00 (b), 12:00 (c) and 18:00 (d)." to

    "Fig. 5 for 00:00, 06:00, 12:00 and 18:00."

44. Page 9, Line 22: change resulted from AC3: Comment 10.

    AuthCM: Changed in text: "Apr" to "April"

45. Page 9, Line 24: change resulted from AC3: Comment 10.

    AuthCM: Changed in text: "Apr" to "April"

46. Page 9, Line 29: change resulted from AC3: Comment 10.

    AuthCM: Changed in text: "Apr" to "April"

47. Page 9, Lines 33–34: change resulted from AC3: Comment 6.

    AuthCM: Changed in text: "for the layers L1 (a), L2 (b), L3 (c) and total column (d)" to

    "for the layers L1, L2, L3 and total column"

48. Page 9, Line 34: change resulted from AC3: Comment 10.

    AuthCM: Changed in text: "Apr" to "April"

49. Page 10, Lines 2–3: change resulted from AC3: Comment 10.

    AuthCM: Changed in text: "Apr" to "April" and "Mar" to "March"

50. Page 10, Lines 6–8: change resulted from AC3: Comment 10.

    AuthCM: Changed in text: "Apr" to "April" and "Mar" to "March"

51. Page 10, Line 13: change resulted from AC3: Comment 10.
    AuthCM: Changed in text: "Apr" to "April"

52. Page 10, Line 16: change resulted from AC3: Comment 10.
    AuthCM: Changed in text: "Apr" to "April"

53. Page 10, Lines 26–27: change resulted from AC3: Comment 10.
    AuthCM: Changed in text: "Apr" to "April" and "Mar" to "March"

54. Page 10, Line 30: change resulted from AC3: Comment 10.
    AuthCM: Changed in text: "Mar" to "March"

55. Page 10, Line 32: change resulted from AC3: Comment 10.
    AuthCM: Changed in text: "Mar" to "March"

56. Page 10, Line 32: change resulted from AC3: Comment 6.
    AuthCM: Changed in text: "(not shown in this paper)" to
    "(not shown)"

57. Page 11, Line 3: change resulted from AC3: Comment 10.
    AuthCM: Changed in text: "Apr" to "April"

58. Page 11, Line 4: change resulted from AC3: Comment 10.
    AuthCM: Changed in text: "Apr" to "April"

59. Page 11, Line 8: change resulted from AC3: Comment 10.
    AuthCM: Changed in text: "Apr" to "April"

60. Page 11, Lines 9–15: change resulted from AC3: Comment 7.
    AuthCM: Changed in text "transverse" to
    "traverse"

61. Page 11, Line 18: change resulted from AC3: Comment 10.
    AuthCM: Changed in text: "Apr" to "April"

62. Page 11, Lines 28–29: change resulted from AC3: Comment 10.
    AuthCM: Changed in text: "Apr" to "April"

63. Page 12, Line 5: change resulted from AC3: Comment 8.
    AuthCM: Changed in text "Garmisch-Partenkirchen, Pillersdorf and Bucharest" to
    "Garmisch-Partenkirchen, Pillersdorf and Bucharest, and"

64. Page 12, Line 6: change resulted from AC3: Comment 8.

AuthCM: Changed in text: "the adsorption of the $SO_2$ on the dust mineral oxides compounds." to "the adsorption of the $SO_2$ on oxides contained in the mineral dust."

65. Page 12, Lines 9–12: change resulted from AC2: Comment 27.

RefC2: Figure 16: "it does not contribute significantly to the discussion. I suggest to exclude this figure, or to move it to the supplementary material."

AuthR: OK

AuthCM: Text was moved to the Supplement of the paper.

66. Page 12, Line 14: change resulted from AC3: Comment 10.

AuthCM: Changed in text: "Apr" to "April"

67. Page 12, Lines 27–30: change resulted from AC1: Comment 4 Part A.

RefC1: "P10 / L27 & 28: - 'No contributions from Europe are seen for these layers.' This may be true for the period in April, as there may not have been any deep convection. However, it would be interesting to also study a summer period with strong convective activity over Central Europe (obviously, in a separate paper !)."

AuthR: The summer periods for the years 2014–2017 are under study, and a paper is under preparation.

AuthCM: Added text: "The spring period studied in this paper is characterized by low, if any, deep convection. For the summer period, one expects however to have a strong convective activity over Central Europe. A study of the summer periods for the years 2014–2017 for the same region was also performed; the results will be presented in a separate paper."

68. Page 13, Lines 10–12: change resulted from AC2: Comment 27.

RefC2: Figure 16: "it does not contribute significantly to the discussion. I suggest to exclude this figure, or to move it to the supplementary material."

AuthR: OK

AuthCM: Text was moved to the Supplement of the paper.

69. Page 19, Figure 2: change resulted from AC3: Comment 10.

AuthCM: Changed in caption: "Apr" to "April" and "Mar" to "March"

70. Page 20, Figure 3: change resulted from AC3: Comment 10.

AuthCM: Changed in caption: "Apr" to "April" and "Mar" to "March"

71. Page 21, Figure 4: change resulted from AC3: Comment 10, from AC1: Comment 5 and from AC2: Comment 19.

AuthCM: Changed in caption: "Apr" to "April"

RefC1: "Figs. 4 & 13 - Key for variables needs to be enlarged P8 / L20 - Height of layers 'amsl' or 'AGL' (also in tables)"

AuthR: The recommended corrections were done in the two figures. AGL was added to text and to the caption for the two figures.

AuthCM: Changed in caption:"Altitude are given in km AGL."

RefC2: Figure 4: "the legend is illegible, text must be enlarged. To better interpret this Figure, it is important to know which profiles correspond to daytime and nighttime (i.e., local time of each plot)."

AuthR: We have improved the figure. We also added in the caption of Fig. 4 the UTC–local time difference.

AuthCM: Changed fonts in the legend of Fig. 4; modified caption: changed "aerosol" to "total aerosol"; added "Local time is UTC+2.".

72. Page 22, Figure 5: change resulted from AC3: Comment 10, from AC3: Comment 11 and from AC2: Comment 20.

AuthCM: Changed in caption: "Apr" to "April"

AuthCM: Changed color of maps to gray (continents) and white (oceans) in the figures.

RefC2: Figure 5: "there are too many lines (altitudes) in the lower plots of sub-figures, it is difficult to interpret. There must be a compromise between completeness and un- derstandability. I suggest that you keep only 3-4 representative altitudes (low, medium, high)."

AuthR: The figure was changed, keeping only the trajectories passing over the lidar stations involved in this analysis.

AuthCM: New Figure 5

73. Page 23, Figure 6: change resulted from AC3: Comment 10, from AC3: Comment 11 and from AC2: Comment 21.

AuthCM: Changed in caption: "Apr" to "April"

AuthCM: Changed color of maps to gray (continents) and white (oceans) in the figure.

RefC2: "Figures 6, 10c, 11c, 12d: you must indicate the locations of the monitoring stations in the maps."

AuthR: The figures were changed adding the location of the monitoring stations.

AuthCM: New Figure 6

74. Page 24, Figure 7: change resulted from AC3: Comment 9 and from AC2: Comment 32.

AuthCM: Changed in caption: "log(range corrected signal)" to

"Logarithm of the range corrected signal"

RefC2: Figure 7: "scales are illegible."

AuthCM: Figure 7 The fonts for scales were increased to be more legible.

75. Page 26, Figure 9: change resulted from AC3: Comment 10, from AC3: Comment 11 and AC2: Comment 34.

AuthCM: Changed in caption: "Apr" to "April"

AuthCM: Changed color of maps to gray (continents) and white (oceans) in the figures.

RefC2: Figure 9: "what are the units for the color map? In addition, the units of longitude should be "degrees", and not "degrees E"."

AuthR: The units for color map are seconds. The units of longitude and latitude were corrected.

AuthCM: Added units for color map and changed units of longitude and latitude in the figures.

76. Pages 27–29, Figures 10, 11, 12: change resulted from AC3: Comment 11, from AC2: Comment 34, from AC2: Comment 21 and from AC2: Comment 22.

    AuthCM: Changed color of maps to gray (continents) and white (oceans) in the figures.

    RefC2: "In addition, the units of longitude should be "degrees", and not "degrees E".."

    AuthR: The units of longitude and latitude were corrected.

    AuthCM: Changed units of longitude and latitude in the figures.

    RefC2: "Figures 6, 10c, 11c, 12d: you must indicate the locations of the monitoring stations in the maps."

    AuthR: The figures were changed adding the location of the monitoring stations.

    AuthCM: New figures 10c, 11c, 12d

    RefC2: Figure 10c: "the model calculates SO2 < 1 ug/m3 for layer 1, which is inside the boundary layer, all over Europe. How does it compare to your ground based measurements?"

    AuthR: There was a bug in superposing the concentration distribution on map for the zoomed distribution (only). The figures 10c, 11c and 12d were replaced with figures with correct data. There is a good agreement between model and ground based measurements, as can be seen from Fig. 1 and Fig. 10c.

    AuthCM: the figures 10c, 11c and 12c were replaced.

77. Page 30, Figure 13: change resulted from AC3: Comment 10 and from AC1: Comment 5.

    AuthCM: Changed in caption: "Apr" to "April"

    RefC1: "Figs. 4 & 13 - Key for variables needs to be enlarged P8 / L20 - Height of layers 'amsl' or 'AGL' (also in tables)"

    AuthR: The recommended corrections were done in the two figures. AGL was added to text and to the caption for the two figures.

    AuthCM: Changed in caption:"Altitude are given in km AGL."

78. Page 31, Figure 14: change resulted from AC3: Comment 10.

    AuthCM: Changed in caption: "Apr" to "April"

79. Page 32, Figure 15: change resulted from AC3: Comment 10, from AC3: Comment 11 and from AC2: Comment 34.

    AuthCM: Changed in caption: "Apr" to "April"

    AuthCM: Changed color of maps to gray (continents) and white (oceans) in the figures.

    RefC2: "what are the units for the color map? In addition, the units of longitude should be "degrees", and not "degrees E"."

    AuthR: The units for color map are seconds. The units of longitude and latitude were corrected.

    AuthCM: Added units for color map and changed units of longitude and latitude in the figures.

80. Page 32, Figure 16: change resulted from AC3: Technical corrections: content moved to Supplement.

AuthCM: Moved Fig. 16 to Supplement.

81. Page 33, Table 1: change resulted from AC3: Comment 10.

AuthCM: Changed in caption: "Apr" to "April"

82. Page 34, Table 2: change resulted from AC3: Comment 10.

AuthCM: Changed in caption: "Apr" to "April"

83. Page 35, Table 3: change resulted from AC3: Comment 10.

AuthCM: Changed in caption: "Apr" to "April"

84. Page 36, Table 4: change resulted from AC3: Comment 10.

AuthCM: Changed in caption: "Apr" to "April"

85. Page 37, Table 5: change resulted from AC3: Comment 10.

AuthCM: Changed in caption: "Apr" to "April"

86. Page 38, Table 6: change resulted from AC3: Comment 10.

AuthCM: Changed in caption: "Apr" to "April"

**Paper supplement**

A paper supplement is added to the revised version. List of added text and figures in the supplement:

1. Page 1: added Figure S1 as resulted from AC2: Comment 8.

RefC2: Page 3, Line 25-28 (Methods): "It is very important to include a map showing the location of all stations explored in this manuscript. That will improve understandability for the readers that are not familiar with EARLINET and with general aspects of Europe geography."

AuthR: OK.

AuthCM: A map has been added to the Supplement.

2. Pages 2–4: added Figure S2, Figure S3, Figure S4 as resulted from AC1: Comment 3 and from AC2: Comment 26.

RefC1: "Since the trajectories in Fig.5 indicate sources processes from almost all over Europe (understandable, especially in the lower & mid levels), but also very distant sources (mostly in elevated layers), the authors should show the relevant meteorological maps for the study period (850, 700, 500, 300 & 200 or 250 hPa circulation) to provide physical evidence for 'conflicting' circulations in some of the layers, and especially for the 'outlying regions'. Of course, FLEXTRA ingests the upper air data from ECMWF, but a cross-verification with 'real meteorological data' will make the cases more convincing."

AuthR: There is no reason to not trust the FLEXTRA calculations. However, seeing weather maps may help to understand the prevailing synoptic pattern. Therefore, a few weather maps are now provided in the Supplement.

AuthCM: Added weather maps as supplement.

RefC2: Page 10: "meteorological maps for the case study period would help to support the conclusions on aerosol transport. Particularly, trajectories calculated below 2000 m are more prone to uncertainties."

AuthR: The meteorological maps will be added to the Supplement of the paper.

AuthCM: meteorological maps in the supplement.

3. Page 5: Moved Figure 16 (GIOVANNI maps) and the corresponding text and aknowledgement from paper to supplement as resulted from AC2: Comment 27.

RefC2: Figure 16: "it does not contribute significantly to the discussion. I suggest to exclude this figure, or to move it to the supplementary material."

AuthR: OK

AuthCM: Fig. 16 was moved to the Supplement of the paper.

[revised manuscript text omitted]